



# Oxygen isotope composition of the final chamber of planktic foraminifera provides evidence of vertical migration and depth-integrated growth

**Hilde Pracht[1], Brett Metcalfe[1,2], and Frank J. C. Peeters[1]**

[1]Earth and Climate Cluster, Department of Earth Sciences, Faculty of Sciences, Vrije Universiteit Amsterdam,
de Boelelaan 1085, 1081 HV, Amsterdam, the Netherlands
[2]Laboratoire des Sciences du Climat et de l'Environnement, LSCE/IPSL, CEA-CNRS-UVSQ, Université Paris-Saclay,
91191 Gif-sur-Yvette, France

**Correspondence:** Brett Metcalfe (b.metcalfe@vu.nl)

**Abstract.** The translation of the original seawater signal (i.e. ambient temperature and $\delta^{18}O_{sw}$) into distinct chambers of a single shell of a foraminifer during calcification can influence our interpretation of surface ocean conditions of the past, when based upon oxygen and carbon stable isotope geochemistry. In this study three different hypotheses were tested to gain more insight into biological and ecological processes that influence the resultant composition of stable isotopes of oxygen ($\delta^{18}O$) in the shells of planktonic foraminifera. These hypotheses were related to the shell size; the differences in isotopic composition between the final chamber and the remaining shell; and the differences between different species. Shells of *Trilobatus sacculifer*, *Globigerinoides ruber* white and *Neogloboquadrina dutertrei* were picked from the top of multi-core GS07-150-24, of modern age, offshore of north-eastern Brazil ($3°46.474'$ S, $37°03.849'$ W) and analysed for single-shell and single-chamber stable isotope analysis. We show that the mean value of $\delta^{18}O$ of the final chambers ($\delta^{18}O_F$) is $0.2‰ \pm 0.4‰$ ($1\sigma$) higher than the mean value $\delta^{18}O$ of the test minus the final chamber ($\delta^{18}O_{<F}$) of *T. sacculifer*. The formation of the final chamber happens at temperatures that are approximately $1°C$ cooler than the chambers formed prior, suggesting both ontogenetic depth migration to deeper water and a potential offset from the surface signal. Furthermore, we show that there is no statistical difference in the $\delta^{18}O_{sacculifer}$ values of shells of three different size classes of *T. sacculifer*, although the pattern between the different size classes indicates depth migration during the life

and growth of *T. sacculifer*. Comparison of vital effect corrected $\delta^{18}O_{shell}$ between *T. sacculifer*, *G. ruber* white and *N. dutertrei* suggests that *G. ruber* has a slightly shallower depth habitat ($\sim$ 90–120 m) compared to the other two species ($\sim$ 100–130 m). Disentangling depth vs. seasonal habitat is complicated given the commonality between isotope values from similar depths but different seasons; for instance, the same average isotope value will have a shallower depth habitat in May than September. Calculation of seasonal-depth habitat was therefore tested. Our results highlight the complicated nature of interpreting oxygen isotopes even for the modern record.

## 1 Introduction

### 1.1 Stable isotope values in foraminifera

The oxygen isotope ratio in the shells of planktonic foraminifera ($\delta^{18}O_{shell}$) is used to reconstruct changes in water properties of the upper water column (e.g. temperature, salinity, stratification) as well aid in palaeoclimatological reconstructions (e.g. defining water mass characteristics, global ice volume). Understanding how this ratio is translated from the ambient environment into the shells of individual foraminifera is therefore important to aid reconstructions and reduce the uncertainty in reconstructed parameters. The $\delta^{18}O_{shell}$ values recorded are a product of the temperature and the isotopic composition of seawater ($\delta^{18}O_{sw}$), it-

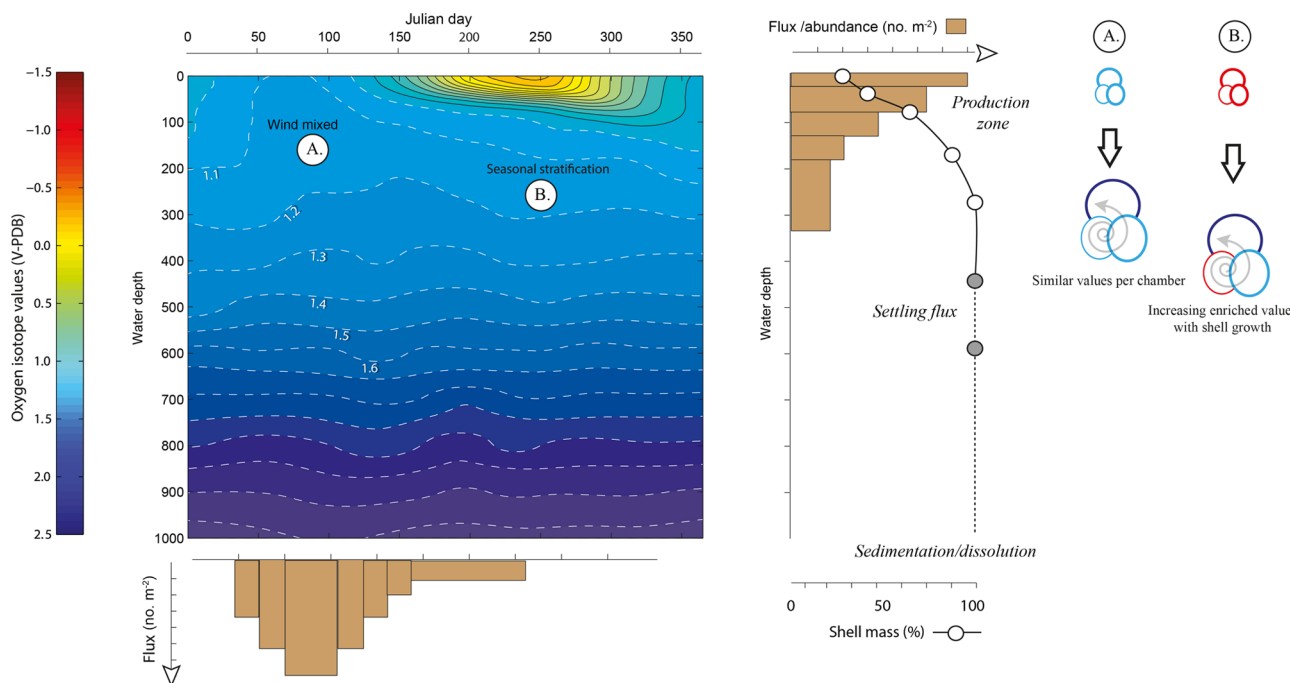

**Figure 1.** Schematic to show the competing individual and population dynamics that may contribute to the variance within the $\delta^{18}$O of a population. Growing seasons may be exaggerated or minimized by water column properties (i.e. rapid population turnover reflecting "bloom" conditions) as highlighted by the uneven widths of the flux histogram. Likewise, rapid growth during juvenile chamber formation may lead to larger offsets (Berger et al., 1978a; Mulitza et al., 1999a) as well as large differences between the surface and deep equilibrium $\delta^{18}$O; however, this is offset by the percentage that these juvenile chambers contribute to the whole shell $\delta^{18}$O. This is particularly influential if scenario B (a stratified water column) occurs rather than scenario A (a well-mixed water column). Schematic modified from Metcalfe et al. (2015).

self a product of the evaporation and dilution (e.g. precipitation, riverine runoff and ice melt) of seawater and hence directly correlated with salinity, which is further modulated by species-specific preferences and metabolic effects (i.e. vital effects). Reconstructions often utilize $\delta^{18}$O produced from a number of pooled specimens, without reconciling how this impacts sample heterogeneity and therefore the resultant climatic interpretation (Fig. 1). Assuming minimal disruption from sedimentary processes such as dissolution (McCorkle et al., 1997) or bioturbation (Hutson, 1980; Lougheed et al., 2018; Löwemark, 2007; Löwemark and Grootes, 2004; Löwemark et al., 2008; Trauth et al., 1997), the variance associated within a pooled $\delta^{18}$O value is a product of the life histories of each individual that comprises the single measurement (Lougheed et al., 2018; Shackleton, 1967) and the underlying biological and ecological controls that govern such "individual" depth distribution within the water column and seasonal occurrence (e.g. Peeters et al., 2002; Schiebel and Hemleben, 2017).

## 1.2 Research question and hypotheses

In this paper we present the results of a number of experiments using single shells and dissected parts of single shells

of planktonic foraminifera. Analysis of small quantities has been made possible with advances in techniques aimed at the routine measurement of microvolume amounts of $CO_2$ (Feldmeijer et al., 2015; Ganssen et al., 2011; Ishimura et al., 2012; Metcalfe et al., 2015; Scussolini et al., 2013; Takagi et al., 2015, 2016; van Sebille et al., 2015; Vetter et al., 2017; Wit et al., 2010, 2013). In order to evaluate the ecological and physiological impacts on the stable isotope values of foraminifera, three species of planktonic foraminifera (*T. sacculifer*; *G. ruber* white and *N. dutertrei*; Fig. 2) were picked from a modern core top sample from the Tropical Atlantic Ocean (Fig. 3). Given its gross morphology, in which individual chambers can be "cleanly" dissected with minimal interference from other chambers (Lougheed et al., 2018; Shuxi and Shackleton, 1989; Spero and Lea, 1993; Takagi et al., 2015, 2016), several experiments were first performed on *T. sacculifer* (Fig. 2i: vii). These experiments focused upon: (1) the differences between successive chambers (Lougheed et al., 2018; Shuxi and Shackleton, 1989; Spero and Lea, 1993; Takagi et al., 2015, 2016); (2) the size–isotope relationship of foraminifera, expanding upon Metcalfe et al. (2015) and Feldmeijer et al. (2015) and (3) the difference in the variance between species. In the section be-

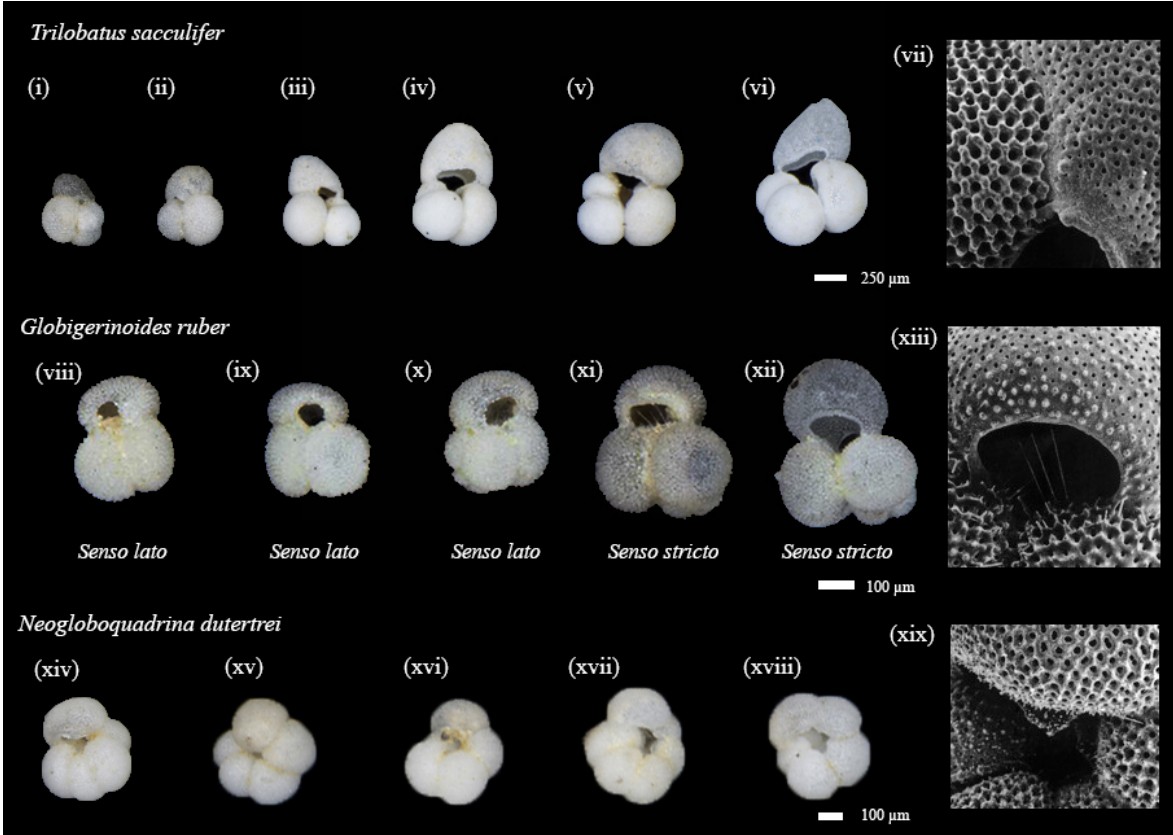

**Figure 2.** Representatives of species used within this study. Light microscope from core top location and scanning electron microscope (SEM) images used to highlight particular features were collected using plankton tows and plankton pumps from the Arabian Sea during the NIOP cruise (Peeters, 2000; Peeters et al., 2002). Note that this final sac-like chamber of *Trilobatus sacculifer* has various unique morphologies, including a thinner walled variety giving the specimen's *F* chamber a translucent quality (similar to i and vi). The species *Globigerinoides ruber* has two morphotypes referred to as (xi–xii) senso stricto (s.s.) and (viii–x) senso lato (s.l.), whilst *Neogloboquadrina dutertrei* is distinguished from other species of *Neogloboquanids* by the presence of a "tooth".

low, we address three fundamental questions related to the oxygen isotope ecology of planktonic foraminifera. In the first question we aim to find out whether there is evidence for depth integrated growth of calcite in a surface-dwelling species. In the second experiment we focus on the question whether shell size and oxygen isotope composition are correlated. Finally, for the third experiment, we investigate whether the oxygen isotope composition of shells of different species from the same geographic location share the same variability.

### 1.2.1 Question 1. Do individuals belonging to the species *T. sacculifer* calcify at one specific depth or undergo depth migration?

The "average" depth habitat of planktonic foraminifera of several species was first defined by Emiliani (1954) revealing that different species occupy discretely different depth habitats, independently corroborated by the later work of Jones (1967) by the presence or absence of species in strati-

fied net tows. However, the offset in $\delta^{18}O$ measured between specimens growing within the euphotic surface waters and those collected from the seabed indicated that depth habitat is not confined to a single depth (Duplessy et al., 1981; Mix, 1987); instead, this "average" species depth habitat would be a weighted average of the various chamber calcification depths occurring during an individual's ontogeny (Kozdon et al., 2009a, b; Shuxi and Shackleton, 1989; Takagi et al., 2015, 2016). Data from plankton tow studies combined with reproduction at depth would suggest that foraminifera migrate through the water column during ontogeny (Fig. 1). For certain species of foraminifera (i.e. *T. sacculifer* and *G. ruber*); however, a portion of the shell may have grown deeper in the water column than the living depths estimated by plankton tows (Lohmann, 1995), i.e. either a calcite crust triggered by temperature change (Hemleben and Spindler, 1983; Hemleben et al., 1985; Srinivasan and Kennett, 1974) or reproduction-triggered gametogenic calcification. For the first objective, we aim to test whether *T. sacculifer* performs depth migration, which would result in a deviation in the geo-

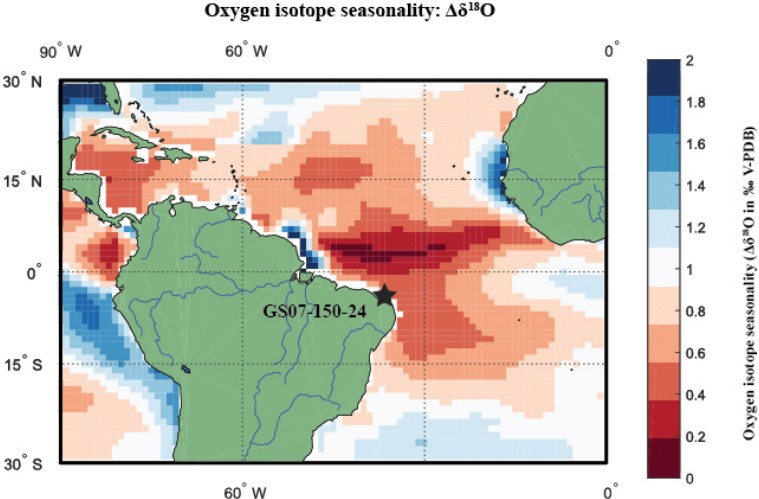

**Figure 3.** TS1 Location Map of RETRO multi-core GS07-150-24 plotted on a basemap of sea surface $\delta^{18}O_{eq}$ seasonality. Location of multi-core (black diamond) plotted on a seasonal oxygen isotope equilibrium ($\Delta\delta^{18}O$) basemap, calculated by subtracting the maximum and the minimum ($\delta^{18}O_{eq}$) of WOA13 temperature and salinity data converted into input variables for a rearranged (Kim and O'Neil, 1997) equation. Core location has an estimated $\Delta\delta^{18}O$ of 0.6 ‰. Note that the coastline basemap of WOA13 is of a far lower resolution than Mathworks MatLab® 2016 Mapping toolbox; thus, white areas around the coast represent a lack of data.

chemistry between the different chambers of a single specimen and also in a deviation from the conditions at the sea surface. A one-sample Student's $t$-test was used to test the claim that there is no difference between the mean of the final chamber and the remaining shell of *T. sacculifer*, i.e. the difference is equal to zero:

Let $X = \delta^{18}O_F - \delta^{18}O_{<F}$.
$H_0 : \mu_X = 0,$
$H_1 : \mu_X \neq 0.$       (1)

By computing the difference and using a reference value of 0, we do not invalidate the rule of independence that a two-sample Student's $t$-test would require between the two sample populations. This dependence is based upon the inference that $\mu_F$ and $\mu_{<F}$ could conceivably be considered to be "before" and "after" measurements, and thus the value of $\mu_{<F}$ could have an impact upon the value of $\mu_F$.

### 1.2.2 Question 2. Does the $\delta^{18}O_{shell}$ of *T. sacculifer* covary with size?

Our second research objective is an expansion of the first objective, as deriving palaeo-SST from the $\delta^{18}O$ compositions of foraminiferal shell is based on the assumption that a given specimen calcifies at, or produces a large proportion of its shell at, one specific depth in the water column. However, a portion of the variability associated with stable isotope measurements in foraminifera is believed to be size-dependent (Ezard et al., 2015). These size dependencies are typically attributed to biological effects and relate to depth migration

through ontogeny (Feldmeijer et al., 2015; Metcalfe et al., 2015). For instance, investigations into the population dynamics of living specimens of *T. sacculifer* in the central Red Sea revealed that whilst this species in general occupies the upper 80 m of the water column distinct size classes were shown to have clear depth preferences (Bijma and Hemleben, 1994; Hemleben and Bijma, 1994) with small foraminifera (100 to 300 µm) in the upper 20 m and progressive larger foraminifera with depth to the point that the largest specimens (> 700 µm) lived between 60 and 80 m. Calcification at different depths throughout their life span may cause a deviation in the $\delta^{18}O$ values of individuals from different sizes, depending on the ambient water column structure, which would therefore reflect different depths and thus the selection of an appropriate size fraction may or may not unduly influence palaeoclimate reconstructions. The aim of this second objective is to further expand upon the results of our first question and test whether the different depth preferences for different sizes of *T. sacculifer*, have an effect on the $\delta^{18}O_{shell}$. Three size fractions were studied to learn more about the effect of size on $\delta^{18}O_{shell}$ and a one-way analysis of variance (one-way ANOVA) with a post hoc test used to detect intra-sample differences was used to test the hypothesis that there is no difference between the means of the different size fractions of *T. sacculifer* TS2:

$H_0 : \mu_{small} = \mu_{medium} = \mu_{large},$
$H_1 :$ at least one of the means is different.    (2)

### 1.2.3 Question 3. Do different species of planktic foraminifera from the same geographic location share the same single-specimen $\delta^{18}O_{shell}$ variability?

Having focused upon a single species for the first two research questions, our third question focuses upon the variability of foraminifera isotope values, which are considered to represent seasonality, and whether fossil shells from different species share similar $\delta^{18}O_{shell}$ variability. Commonly when referencing seasonality, temperature is considered as the variable of interest. However, the tropics have relatively small seasonal temperature variability compared with higher latitudes, the core is situated along the north-eastern coast of Brazil which may be influenced by the shift in the ITCZ (Jaeschke et al., 2007). Temperature and salinity have opposing effects on the overall oxygen isotope composition. Surface-dwelling species of planktonic foraminifera, *T. sacculifer* (Fig. 2i: vii); *G. ruber* (Fig. 2viii: xiii); and the thermocline dwelling *N. dutertrei* (Fig. 2xiv: xix) were picked from the core top. All species are symbiotic (Schiebel and Hemleben, 2017) which limits the depth of the maximum growth. A one-way ANOVA was used to test, whether the means of each species are equal or whether the alternative hypothesis that one or more of the species means differs from one another, with the following hypothesis TS3:

$$H_0 : \overline{\mu_{T.sacculifer}} = \overline{\mu_{G.ruber}} = \overline{\mu_{N.dutertrei}},$$
$$H_1 : \overline{\mu_{T.sacculifer}} \neq \overline{\mu_{G.ruber}} \neq \overline{\mu_{N.dutertrei}}. \quad (3)$$

In addition to the ANOVA test for testing multiple means, we use a Kolmogorov–Smirnov (K–S) test to test whether the species stem from a similar $\delta^{18}O$ distribution, with the claim in the null hypothesis being equal (i.e. not significantly different) distributions. Three tests were carried out: *T. sacculifer* vs. *N. dutertrei*, *T. sacculifer* vs. *G. ruber* and *N. dutertrei* vs. *G. ruber* respectively. For Objective 3, the hypothesis is that the $\delta^{18}O_{shell}$ variability varies for different species from the same location, which would mean that different species from the same location can give a different temperature and/or seasonality derivation (Mix, 1987; Roche et al., 2018). Overall, the hypothesis is that different processes cause deviations from the sea surface equilibrium. More insight into the presence and size of these deviations can possibly be used to account for future climate reconstructions.

## 2 Method and material

### 2.1 Material and general methodology

Multi-core GS07-150-24 was collected on board the RV *G.O. Sars* at a depth of 2412 m offshore of north-eastern Brazil ($3°46.474'$ S, $37°03.849'$ W; Fig. 3). Following sub-sampling, the top of the core was washed over a $> 63\,\mu m$ sieve, dried overnight, before being dry sieved over a 150 and 500 μm mesh. Regardless of the research question, each specimen underwent the same methodological protocol which aims to reduce uncertainty (e.g. specimen misidentification; anomalous or abnormal features) within single-shell stable isotope analysis by cataloguing morphology and physical features of specimens prior to destructive analysis. After picking, the selected specimens were given a unique identifier, imaged in the umbilical position (Fig. 2) using a Nikon Digital Research microscope with a prior motorized stage. The motorized stage enables multiple images to be taken at pre-determined intervals in μm. These images were then combined using Nikon Digital Research D software into an extended depth of focus (EDF) image. Each EDF image was then used to measure the diameter and surface area of both the final chamber and the whole shell, using the same programme. Groups of specimens were imaged together, with little impact upon the resolution (1 pixel, depending on the magnification, is equal to 0.3 to 1.5 μm) and placed into individual slides in order to generate a high throughput. After imaging, specimens were weighed individually in tin capsules using a Mettler-Toledo UMT microbalance (manufacturers precision 0.1 μg). In total 207 specimens of *T. sacculifer* were picked, weighed and measured for size. Following these measurements, specimens selected for research questions 1 ($\delta^{18}O$ difference between $F$ and $< F$) and 2 ($\delta^{18}O$ difference between size) underwent additional steps, outlined in Sect. 2.2 (dissection of chambers) and Sect. 2.3 (size fractions), prior to stable isotope analysis.

For $\delta^{18}O$ and $\delta^{13}C$ analysis, shells and/or single chambers between 5 and 70 μg were placed in a 4.5 mL borosilicate exetainer vial, whereas shells between 20 and 145 μg were placed in larger 12 mL borosilicate exetainer vials (Breitenbach and Bernasconi, 2011; Feldmeijer et al., 2015; Metcalfe et al., 2015). Each vial was sealed with a cap with a pierce able septum, placed in a heated block (45 °C), before being flushed with helium for 3 or 5 min to remove the ambient air (flow rate $> 100\,mL\,min^{-1}$) depending on the size of the vial. Each sample was reacted with a few drops of phosphoric acid ($H_3PO_4$) for 160 min, transferred using a continuous flow of helium into a GasBench II preparation device, in which impurities were removed, before being introduced into a Thermo Delta$^+$ mass spectrometer. Results were reported as $\delta$ values in per mil (‰), following voltage correction of the amplitude of mass 44 using grains of 150–180 μm of Vrije Universiteit Internal Carbonate Standard (VICS: $\delta^{18}O = -5.44\,‰$; $\delta^{13}C = 1.35\,‰$) in order to be placed on the V-PDB scale. The precision of within-run international standards of IAEA-CO-1 and IAEA-CO-603 (minimum $n = 10$), placed to book-end every 6 samples, was better than 0.14 ‰ for both $\delta^{18}O$ and $\delta^{13}C$.

## 2.2 Specific methodology for Question 1

To make inferences about depth migration (Research Question 1) 57 specimens of *T. sacculifer* were picked from two size fractions: 150–500 and $> 500 \, \mu$m. Selection of specimens was based on the following criteria: (1) specimens were intact, or did not appear externally to be broken or damaged; (2) specimens were not visibly discoloured or overly contaminated with clay; (3) specimens were not kummerform (Bé et al., 1971; Berger, 1969, 1970; Olsson, 1973), and/or it was possible for the sac-like chamber to be dissected; and (4) specimens and their final chambers were judged to be heavier than $6 \, \mu$g to ensure sufficient mass for measuring on the mass spectrometer. Following the standard protocol, the final sac-like chamber was amputated (Shuxi and Shackleton, 1989; Spero et al., 1993; Ishimuru et al., 2012) from the rest of the shell with a number 7 dissecting scalpel, so that each shell was analysed in two portions, the last chamber ($\delta^{18}O_F$) and a shell without the last chamber ($\delta^{18}O_{<F}$). Those shells, minus the $F$-chamber, that still exceeded $> 150 \, \mu$g were analysed in two parts. The remainder of the shell was placed between two glass slides, crushed, homogenized and then separated into two portions (identified as A and B). The isotope value of $\delta^{18}O_{<F}$ was calculated by using a weighted mean of the measured $\delta^{18}O$ from these two portions (a and b), with the following:

$$\delta^{18}O_{\mu<F} = \frac{\left(\delta^{18}O_{<F}{}^a \cdot \text{amplitude}^a\right) + \left(\delta^{18}O_{<F}{}^b \cdot \text{amplitude}^b\right)}{\left(\text{amplitude}^a \cdot \text{amplitude}^b\right)}, \quad (4)$$

where the amplitude is the amount of $CO_2$ of mass 44 produced in mVolts, which is linearly related to sample weight.

## 2.3 Specific methodology for Question 2

To make inferences about the effect of size on the measured isotopic composition (*Research Question 2*), 41 whole shells of *T. sacculifer* were picked from the $> 150$ and $> 500 \, \mu$m size fractions and subdivided based upon measured size. Three size classes were determined: small: 222–316 $\mu$m ($n = 10$); medium: 373–467 $\mu$m ($n = 16$); and large: 511–597 $\mu$m ($n = 15$). The size classes have uneven widths with ranges of 94 $\mu$m, 94 and 86 $\mu$m respectively. All $\delta^{18}O_{shell}$ values of *G. ruber* and *T. sacculifer*, irrespective of size, were corrected for their vital effect.

## 2.4 Specific methodology for Question 3

To determine whether species of planktonic foraminifera from the same geographic location share the same or similar single specimens, $\delta^{18}O_{shell}$ variability specimens of *G. ruber* ($n = 20$) and *N. dutertrei* ($n = 14$) were picked from the same interval. These shells underwent the same methodology outlined in Sect. 2.1 for photographing, weighing and isotope analysis.

## 2.5 Atlas data (temperature, salinity and $\delta^{18}O_C$)

World Ocean Atlas 2013 (WOA13; Boyer et al., 2013) was used as an average climatology at the core site, temperature and salinity was extracted from the live access server (LAS) of NOAA. The oxygen isotope equilibrium values calculated by first computing the oxygen isotope of seawater ($\delta^{18}O_{sw}$) from WOA 13 salinity using the oxygen isotope database of LeGrande and Schmidt (2006). A regional mask was used on a global grid to define which regional equation to use, regions were redefined to fit established conventions on the definitions of particular ocean basins (similar to the approach of Roche et al., 2018). Values of salinity that represent riverine outflow (PSU $< 10$) were excluded from the resultant reanalysis of the salinity vs. oxygen isotope of seawater relationship of the tropical Atlantic Ocean (LeGrande and Schmidt, 2006). Both WOA13 temperature and the computed $\delta^{18}O_{sw}$ were then used as input values for the equation of Kim and O'Neil (1997), rearranged from the relationship between temperature and the fractionation of oxygen isotopes in planktonic foraminifera, to derive the oxygen isotope equilibrium ($\delta^{18}O_{eq}$):

$$\delta^{18}O_{eq} = 25.778 - 3.333 \times (43.704 + T)^{0.5} + \delta^{18}O_{sw}, \quad (5)$$

Here, Kim and O'Neil (1997) is used to define an inorganic equilibrium value of $\delta^{18}O_c$ this equation is chosen to avoid potential differences due to (1) light level; (2) foraminiferal size; (3) ontogenetic level (Bemis et al., 1998, 2000); and (4) species (Mulitza et al., 1999b). To account for similar absolute measured values between species which are not produced by concurrent depth or seasonal preferences between species but instead by species-specific disequilibria from values obtained from ambient seawater equilibrium (so-called "vital effects") a correction was applied. The $\delta^{18}O$ values of *T. sacculifer* and *G. ruber* were corrected by $0.48\%$ ($1\sigma = 0.15\%$; Peeters, 2000; Peeters et al., 2004). To understand the results, a probabilistic determination of the seasonal-depth distribution using a fitted normal distribution to the single-specimen data was calculated by fitting the probabilities of $\delta^{18}O_{shell}$ to the seasonal and depth distribution of $\delta^{18}O_{eq}$. Fitting was accomplished using a normal distribution; therefore, to test whether the data come from a normal distribution, a K–S test (data normalized first) and an Anderson–Darling test were performed. The probability determined for each $\delta^{18}O_{shell}$ is then transposed onto the $\delta^{18}O_{eq}$ of the core top.

## 3 Results

To aid the reader, the stable isotope values in the following section are reported to two or three decimal places to report the results of the statistical tests without introducing rounding errors; this should not be misconstrued as reflecting a

greater degree of certainty in the isotope values after the decimal point.

## 3.1 Size vs. weight of *T. sacculifer*

During the picking and selection process, a total of 207 specimens of *T. sacculifer* was measured and weighted. Ninety-eight of these were eventually analysed for stable isotopes; however data on size and weight for all 207 specimens were processed to make interferences about the relation between these two parameters (Fig. 4). For comparison, the measured size and weight data were plotted alongside a theoretical hollow foraminifer (similar to *Orbulina universa*) in which the shell weight is calculated by assuming a constant porosity and the density of calcite is $2.71 \, \mathrm{kg^{-1} \, m^{-2}}$. This approach highlights the complexity when dealing with foraminiferal weight when both chamber number and chamber wall thickness is variable, there is a clear increase in the spread in shell weight (Fig. 4a) when the area is larger than $4 \times 10^5 \, \mathrm{\mu m^2}$ this is likely either the result of chamber thickening or non-linear growth of the foraminiferal shell. After completion of the first chamber, during the construction of subsequent chambers of a Rotaliid foraminifer an additional layer of calcite is added to the previous chambers, making them a incrementally thicker. This makes the weight increase deviate from a linear relation and also makes that the final chamber has less thick (and therefore lighter) walls than its predecessors (Bé and Lott, 1964). Regarding shell size vs. shell weight, a heteroscedastic relation was found. For smaller tests, little variance was present, deviation from the regression line increased when the area of the test increased, indicating more variability in shell weight for bigger shells. A possible explanation can be found in the fact that when shells grow larger, they tend to get more divergent or erratic forms (Fig. 4a), this especially goes for the final chambers. A relatively low weight in large specimens is then caused by a relatively large $F$ chamber. Because the $F$ chamber has a relatively thin wall and therefore a low weight, the shell of large specimens is lighter than expected (Bé and Lott, 1964). A relatively high weight in large specimens is caused by a big $< F$ and small $F$. The chambers of $< F$ have thicker walls and therefore a relatively high weight, causing a positive deviation from the size-weight regression line. A heteroscedastic relation also appeared between the area of the final chamber and the area of the whole shell. In Fig. 4b it is visible that when the area of the whole shell size increases, the variance becomes bigger. Large specimens often have disproportionally large or small final chambers, with no clear relationship between total shell size and the size of the final chamber. Compared to smaller shells, big shells tend to have relative small or big final chambers, which are not in proportion with the shell.

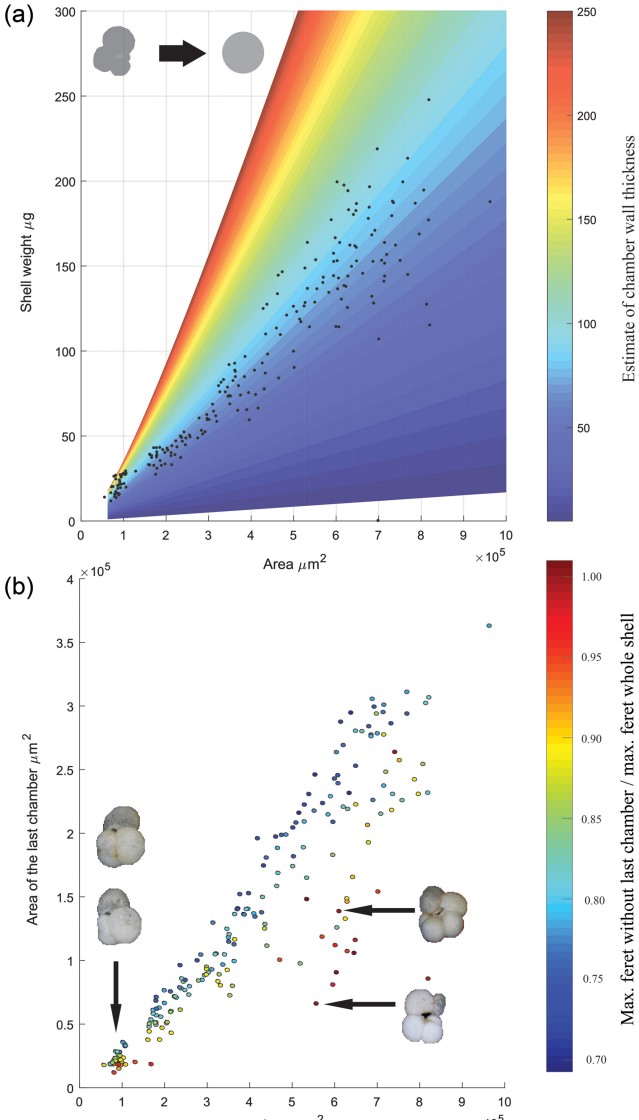

**Figure 4.** Physical properties of *T. sacculifer*. **(a)** Size vs. weight of *T. sacculifer*; the data are overlaid on a theoretical calculation of what the shell weight would be of a spherical hollow foraminifer with consistent porosity, assuming the density of calcite. Colour represents variation in wall thickness used to calculate the difference between the inner and outer sphere volume. **(b)** The area of the final chamber vs. the whole shell area of specimens measured, pictures inset highlight the morphology associated with the spread in the datasets. The scatter points filled colour reflects the ratio between the maximum ferret diameter with and without the last chamber. The linear regression equation is $y = 0.0002x - 4.5243$ ($r^2 = 0.8985$; $n = 207$).

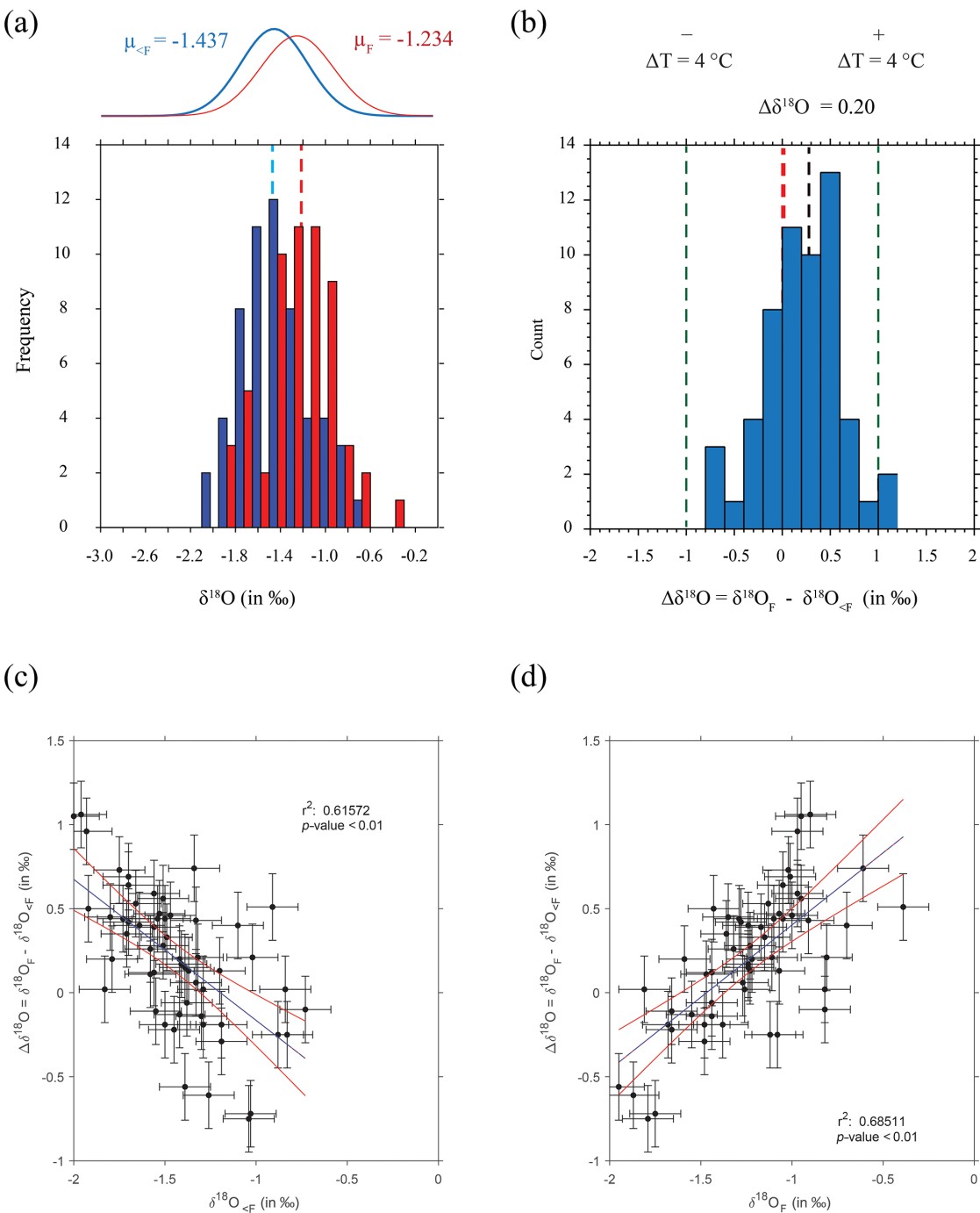

**Figure 5.** TS4 Final chamber vs. rest of shell $\delta^{18}O$. **(a)** Raw $\delta^{18}O$ values of rest of shell (blue) and final chamber (red) plotted as a histogram, vertical bars represent sample means. Fitted normal distributions for rest of shell ($\mu < F$; blue) and final chamber ($\mu F$ red), with mean average values are also indicated. Histogram bins are in 0.25 ‰ bin intervals, the equivalent of $\sim 1\,°C$ depending on whether the data are on the high or low end of the scale. **(b)** Histogram of the same specimen difference in $\delta^{18}O$ ($\equiv \Delta\delta^{18}O$) between $< F$ and $F$. Vertical green lines at $\Delta\delta^{18}O$ of $\pm 1$ ‰ represent a $\sim 4\,°C$ temperature variation, a vertical red line denotes no difference ($\Delta\delta^{18}O = 0$). The average $\Delta\delta^{18}O$ ($\mu = 0.203$) is shown as a black vertical line. **(c, d)** Scatter plot of the same data plotted as either the **(c)** $< F$ remaining of the shell and **(d)** the $F$ chamber $\delta^{18}O$ vs. $\Delta\delta^{18}O$. Vertical error bars represent the square root of the sum of the error's squared $\left( \sqrt{\left( \text{Error A}^2 + \text{Error B}^2 \right)} \right)$ and the horizontal a single machine error. Blue line is the linear regression and the blue a confidence interval on the regression. All values in per mil (‰) on the V-PDB scale.

**Table 1.** Results of the Student's $t$-test for a single group. Testing the $\delta^{18}O_F$-$\delta^{18}O_{\text{shell-}F}$. Test value: 0 (testing whether the difference between $\delta^{18}O_F$ and $\delta^{18}O_{\text{shell-}F}$ is statistically equal to 0).

| Count | 57 |
|---|---|
| Mean | 0.20 |
| Variance | 0.16 |
| SD | 0.40 |
| Std. error | 0.054 |
| Mean difference | 0.20 |
| Degrees of freedom | 56 |
| $t$ value | 3.78 |
| $t$ probability | 0.0004 |

## 3.2 Question 1: depth migration

Measured $\delta^{18}O$ values are plotted as a histogram in Fig. 5a for $\delta^{18}O_{<F}$ and $\delta^{18}O_F$. The mean $\delta^{18}O$ values for the final chambers and the shells without final chambers, were [TS5] $(\overline{\delta^{18}O_F} = \mu_F =) -1.234\,\text{‰}$ and $(\overline{\delta^{18}O_{<F}} = \mu_{<F} =) -1.437\,\text{‰}$ respectively, indicating the means of the two groups differ by approximately $0.203\,\text{‰}$, with the final chamber having a more negative value than the shells without the final chamber. In Fig. 5b, a histogram of $\Delta\delta^{18}O$, which represents the difference between $\delta^{18}O_{<F}$ and $\delta^{18}O_F$ is shown. The data are normally distributed, with a mean (difference) of $+0.23\,\text{‰}$ (Table 1). The one-sample $t$-test results in a $p$-value of $< 0.05$, therefore the null hypothesis can be rejected at a significance level of $\alpha = 0.05$, and it is possible to conclude that the $\Delta\delta^{18}O$ is statistically different from 0. In other words, the difference between final chamber $\delta^{18}O$ and the $\delta^{18}O$ value of the shell with the last chamber removed, is positive and significantly different from zero at the 95 % confidence level. The positive value indicates that growth of the final chamber occurs, on average, at a lower temperature. A scatter plot between $\Delta\delta^{18}O$ and $\delta^{18}O_{<F}$ ($r = 0.61$; $n = 57$); and $\delta^{18}O_F$ ($r = 0.69$; $n = 57$) shows that there is a statistically significance between the variables (Fig. 5c and d). The significance of $r$ being non-zero was statistically tested using Pearson's correlation coefficient. The critical value for the absolute value of the correlation coefficient for an $\alpha$ of 0.05 where $n = 50$ is 0.273 and $n = 60$ is 0.250. Our $n$ ($= 57$) taking into account the number of degrees of freedom (d.f. $= n - 2$) lies between these values of $n$. Since our correlation coefficients are higher than these critical values, it is possible to conclude that they are different from zero. The correlation coefficients are also significant for an $\alpha$ of 0.01 (C.V. $= 0.354$; for d.f. $= 50$).

## 3.3 Question 2: covariance with size

The mean $\delta^{18}O$ for the small, medium and large size fractions of *T. sacculifer* was $-1.12\,\text{‰}$, $-1.30\,\text{‰}$ and $-1.15\,\text{‰}$ respectively (Fig. 6; Table 2), with the smallest and largest shells having a less negative mean value than the medium shells. The resultant ANOVA test $p$-value of 0.136 ($> 0.05$) however indicates that the null hypothesis of equal means cannot be rejected; the observed differences between the different size classes are therefore not enough to state that there is a statistically significant difference between the mean $\delta^{18}O_{\text{shell}}$ of the small ($222$–$316\,\mu\text{m}$); medium ($373$–$467\,\mu\text{m}$) and large ($511$–$597\,\mu\text{m}$) shells.

## 3.4 Question 3: similarity in species-specific variability for a single site?

The mean $\delta^{18}O$ of the single specimens of *N. dutertrei*, *T. sacculifer* and *G. ruber* were $-0.84\,\text{‰}$; $-0.82\,\text{‰}$ and $-1.15\,\text{‰}$ respectively (Fig. 6b; Table 3). An ANOVA to test whether the species had equal means resulted in a $p$-value of 0.0001 which led to a rejection of the null hypothesis ($p < 0.05$) that the species have equal means. A post hoc Tukey all pair comparison, using vital effect corrected $\delta^{18}O$ values, shows that the mean $\delta^{18}O_{\text{shell}}$ of *G. ruber* differed significantly from both *T. sacculifer* ($p = 0.0004$) and *N. dutertrei* ($p = 0.0017$), whereas the difference between *T. sacculifer* and *N. dutertrei* was not significant ($p = 0.9492$). Using the uncorrected, for vital effect (Table 4), $\delta^{18}O$ values all species show statistical difference between one another. The range in species $\delta^{18}O$ is less than $1\,\text{‰}$, from largest to smallest the range of *N. dutertrei* ($\delta^{18}O_{\text{min.}}$: $-1.33\,\text{‰}$; $\delta^{18}O_{\text{max.}}$: $-0.46\,\text{‰}$; $\delta^{18}O_{\text{range}}$: $0.86\,\text{‰}$) is larger than *T. sacculifer* (min. $-1.20$; max. $-0.39$; range: $0.81\,\%$) and *G. ruber* (min. $-1.55$; max. $-0.78$; range $0.76$) (Fig. 6b). However, for three $F$-tests, to determine whether the species has equal variances, the resultant $F$-value is less than the $F$-test critical value and therefore the null hypothesis that they have equal variances could not be convincingly rejected with the data measured. A Kolmogrov–Smirnov test was used to test whether the three species come from the same distribution, which would indicate the three species have recorded the same climate signal. Three tests were carried out, each comparing the distributions of two species at the time. The test comparing *T. sacculifer* and *N. dutertrei*, resulted in a $p$-value of 0.9697 ($\alpha = 0.05$), meaning it is unable to reject the null hypothesis. This means that we found no evidence that the two species have a different probability distribution. For *N. dutertrei* vs. *G. ruber* ($p = 0.012$) and *T sacculifer* vs. *G. ruber* ($p = 0.030$), the null hypothesis however could be rejected; therefore, the two species record significantly different variability in $\delta^{18}O$.

## 4 Discussion

## 4.1 Depth migration

Numerous studies have subdivided the species *T. sacculifer* into the forms with a distinct final chamber, referred to as "sac-like", from "non-sac" forms referred to by its junior

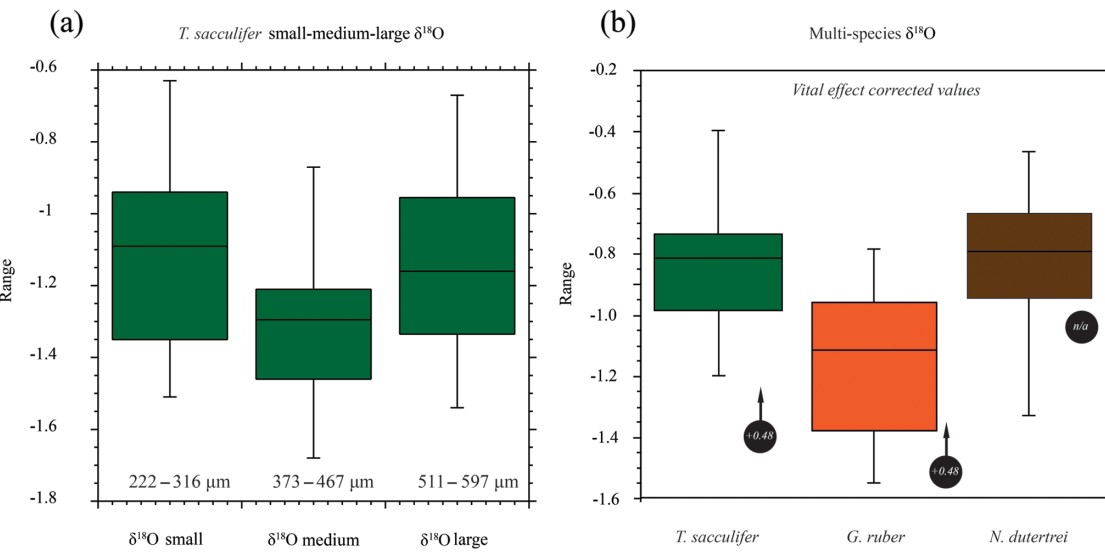

**Figure 6.** Box plots of the oxygen isotope values ($\delta^{18}O$) vs. size and for different species of planktonic foraminifera. **(a)** Small, medium and large (see the text for definitions of sizes represented; note the uneven size interval). **(b)** The (right-hand side) box plots are the range in oxygen isotopes corrected for vital effects (circles with arrows) calculated from in situ water sampling (plankton pump and plankton tow; Frank Peeters, unpublished data). The central bar of each box represents the median.

**Table 2.** Results of the ANOVA test for size fractions. Comparison of the means of the three different size classes: small, medium and large.

| Analysis of variance results | | | | | |
|---|---|---|---|---|---|
| Source | Degrees of freedom (DF) | Sum of squares (SS) | MS | $F$-value | $P$-value |
| Total | 40 | 2.49 | 0.06 | 2.12 | 0.13 |
| A | 2 | 0.25 | 0.13 | | |
| Error | 38 | 2.24 | 0.06 | | |

| Tukey's all pair comparison TS6 | | | | |
|---|---|---|---|---|
| | Mean difference | $\|q\|$ | $P$ | 95 % CL |
| $\delta^{18}O_{shell}$ small vs. $\delta^{18}O_{shell}$ medium | 0.17 | 2.52 | 0.19 | −0.06 to 0.41 |
| $\delta^{18}O_{shell}$ small vs. $\delta^{18}O_{shell}$ large | 0.03 | 0.36 | 0.9646 | −0.22 to 0.27 |
| $\delta^{18}O_{shell}$ large vs. $\delta^{18}O_{shell}$ medium | 0.15 | 2.41 | 0.2162 | −0.06 to 0.36 |

synonym *G. trilobus*. The division between these forms is not exclusively for studies with geochemical analysis but is also commonly found in studies using faunal abundance counts. In fact, a number of studies where *T. sacculifer* is used as a proxy for palaeoclimate have removed the final chamber to avoid potential bias caused by the assumed depth migration (Coadic et al., 2013). In our results we show a mean difference of approximately 0.203‰ between $\delta^{18}O_F$ and $\delta^{18}O_{<F}$, i.e. those forms that would be described as *T. sacculifer* and those as *G. trilobus*, with $< F$ having more negative values TS7 ($\mu = -1.437$‰) than $F$ ($\mu = -1.234$‰).

A number of species of foraminifera, including the species analysed here (Bird et al., 2018), are associated with symbiotic algae that undergo diurnal migration into and out of the shell and vacuoles in the foraminifer's cytoplasm, a ma-

jor function is their facilitation of both growth and longevity of an individual foraminifer (Anderson and Be, 1976; Bé et al., 1982; Caron et al., 1982; Faber et al., 1988, 1989; Gastrich, 1987; Hemleben et al., 1989; Spero and DeNiro, 1987; Spero and Lea, 1993; Spero and Parker, 1985). As such the presence of symbionts places limits upon the range of depth habitat: juvenile foraminifer must either be re-infected by or capture new symbiotic algae (Hemleben et al., 1989; Spero, 1998) whilst adult foraminifera with symbiotic associations would do well to remain within the photic zone. Using the mean $\delta^{18}O_{sw}$ of the sample location of 0.42‰ (WOA13; Boyer et al., 2013) and the mean $\delta^{18}O$ of $F$ and $< F$ respectively, mean temperatures of 24.5 and 25.5 °C were derived which indicates a potential mean depth below and above 100 m respectively. The euphotic zone depth varies both re-

**Table 3.** Results of the ANOVA test comparing the means of the three different species corrected for vital effect: *T. sacculifer*, *G. ruber* and *N. dutertrei*.

| Analysis of variance results | | | | | |
|---|---|---|---|---|---|
| Source | DF | SS | MS | *F* | *P* |
| Total | 49 | 3.85 | 0.08 | | |
| A | 2 | 1.22 | 0.61 | 10.91 | 0.00013 |
| Error | 47 | 2.63 | 0.06 | | |

| Tukey's all pair comparison | | | | |
|---|---|---|---|---|
| | Mean difference | $|q|$ | *P* | 95 % CL |
| *T. sacculifer* vs. *G. ruber* | 0.33 | 5.90 | 0.0004 | 0.14 to 0.52 |
| *T. sacculifer* vs. *N. dutertrei* | 0.03 | 0.44 | 0.9492 | −0.18 to 0.24 |
| *N. dutertrei* vs. *G. ruber* | 0.30 | 5.22 | 0.0017 | 0.104 to 0.504 |

**Table 4.** Results of the ANOVA test comparing the means of the three different species uncorrected for vital effect: *T. sacculifer*, *G. ruber* and *N. dutertrei*.

| Analysis of variance results | | | | | |
|---|---|---|---|---|---|
| Source | DF | SS | MS | *F* | *P* |
| Total | 49 | 7.71 | 0.16 | | |
| A | 2 | 5.07 | 2.53 | 44.99 | < 0.0001 |
| Error | 47 | 2.65 | 0.06 | | |

| Tukey's all pair comparison | | | | |
|---|---|---|---|---|
| | Mean difference | $|q|$ | *P* | 95 % CL |
| *T. sacculifer* vs. *G. ruber* | 0.78 | 13.41 | < 0.0001 | 0.58 to 0.98 |
| *T. sacculifer* vs. *N. dutertrei* | 0.45 | 7.39 | < 0.0001 | 0.24 to 0.66 |
| *N. dutertrei* vs. *G. ruber* | 0.33 | 5.87 | 0.0004 | 0.14 to 0.52 |

gionally and temporally, from a lower limit of 20 m to greater than 120 m globally, with measured sites displaying variability between < 40 to > 100 m on seasonal timescales (Buesseler and Boyd, 2009; Siegel et al., 2014). Spero (1998) reflecting on the evolutionary advantages for a species known to harbour symbiotic algae to calcify below the photic zone considered that there are none, and instead as planktonic foraminifera are at the mercy of ocean currents such specimens that reflect too deep growth (Lohmann, 1995) could represent descent or advection out of their suitable habitat range. In fact, our results highlight the complexity of the individual life histories of individual foraminifer like many species of (phyto- and or zoo-)plankton which are heavier than water (Huisman et al., 2002) their persistence within the upper water column, despite a sinking trajectory that should take them below conditions of light and nutrients sufficient for growth, may relate to turbulence and advection (Huisman et al., 2002; Margalef, 1978; Riley et al., 1949; Shigesada and Okubo, 1981; Sverdrup, 1953). Our results show that whilst the mean difference in $\delta^{18}$O, between $F$ and $< F$, is weighted toward a colder signal within the final chamber,

there are however a number of shells that record a warmer signal in the $F$ chamber ($n = 8$ for $< -0.25$ ‰; or $n = 16$ for $< 0.0$ ‰). Although the role of turbulence remains enigmatic (Davila and Hunt, 2001; Ruiz et al., 2004), with Margalef (1978) suggesting that favoured species (i.e. those with spines or bubble capsules) and size of specimens depend on whether turbulence is low or high, within a turbulent water column the overall population average may suggest a trajectory of a downward descent, whereas the descent of an individual shell may be much more complicated. Our results suggest that there is a difference between chambers $F$ and $< F$; on average the formation of the final chamber occurs in water approximately 1 °C colder than the chambers formed prior, suggesting both ontogenetic depth migration to deeper waters and a potential offset from the surface signal.

The statistical significance between either chambers $F$ $\delta^{18}$O and/or $< F \delta^{18}$O, and the $\Delta \delta^{18}$O (Fig. 5c and d) could indicate that the environment in which the early chambers ($< F$) form determines the final chambers $\delta^{18}$O (Fig. 5c); the warmer SST (more negative $\delta^{18}$O values) specimens have a larger $\Delta \delta^{18}$O, which could indicate the specimen lived dur-

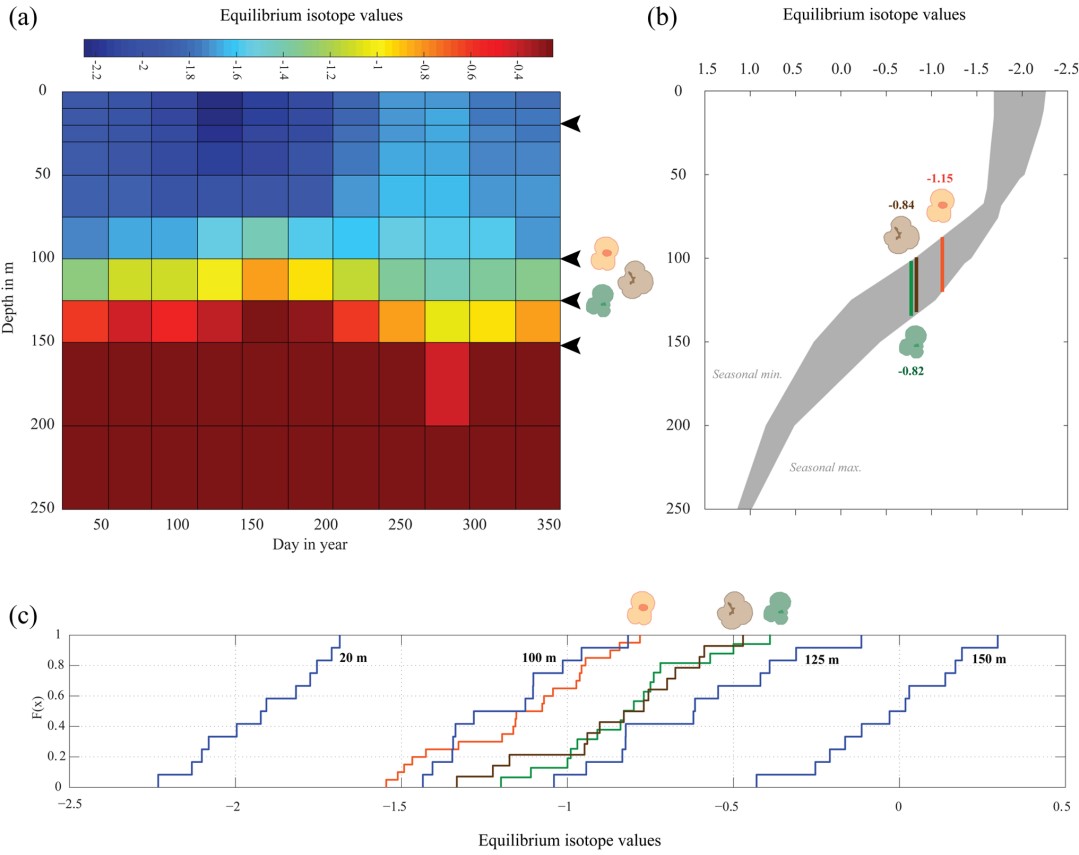

**Figure 7.** Season or depth, predicting likely depth habitats of planktonic foraminiferal species using inferred equilibrium oxygen isotope values ($\delta^{18}O_{eq}$). World Ocean Atlas (WOA13) temperature and salinity were used to compute the equilibrium oxygen isotope value using the tropical Atlantic $\delta^{18}O$–salinity relationship defined by LeGrande and Schmidt (2006) input into a rearranged form of Kim and O'Neil (1997) (see Sect. 2.4). **(a)** Contour plot of $\delta^{18}O_{eq}$, plotted as depth in metres vs. day in year. Note the uneven depth interval distribution inherent within the WOA dataset. Black arrows represent depths chosen in **(c)** to calculate cumulative distribution, species symbols represent inferred depths of mean average values. **(b)** The seasonal minimum and maximum (grey band) in $\delta^{18}O_{eq}$ for each depth interval, coloured bars represent depth intervals that are similar to isotopic values of the three species. **(c)** Cumulative density distribution of the three species plotted alongside the CDF for 20, 100, 125 and 150 m distributions; values are plotted as probability ($F(x)$) vs. oxygen isotope value. All values plotted in per mil (‰) on the V-PDB scale.

ing stratified water conditions (Fig. 1, scenario B). Likewise, the colder SST (more positive $\delta^{18}O$ values) specimens have a smaller difference; therefore, these specimens could represent those that live under mixed conditions (Fig. 1, scenario A). Specimens that show a warming between $F$ and $< F$ chambers could theoretically have calcified during a period of change, a transition from a stratified to a mixed (or vice versa) water column.

## 4.2    Difference between $F$ and $< F$: an underestimation?

The difference in the isotopic value between successive chambers may not depend solely on depth migration during ontogeny but may be altered by chamber thickening. Two types of chamber thickening are known to exist: a calcite crust seen in Neogloboquanids and Globorotalids; and ga-

metogenetic calcite (GAM) seen in *Orbulina universa* and *T. sacculifer*. Whilst both types are produced at the end of the life cycle and therefore deeper in the water column, one is considered to represent low temperature thickening of the shell, and the other a pre-reproduction thickening of the shell. Thickening of the pre-existing chambers that compose a single shell in response to a particular environmental parameter or at the end of the life cycle may bias the resultant isotopic composition; depending on the water column structure and depth of the mixed layer (Fig. 1, scenarios A and B), the calcite produced in such a way may be indistinguishable isotopically from older chambers. Whilst the size of this bias induced by GAM may have been overestimated in the literature, for instance using cultures, Hamilton et al. (2008) showed that approximately 80 % of the shell material is pre-GAM; new evidence suggests that the $\Delta\delta^{18}O$ between pre-GAM and GAM is $\sim 1\%_0$ (Wycech et al., 2018 TS8). The

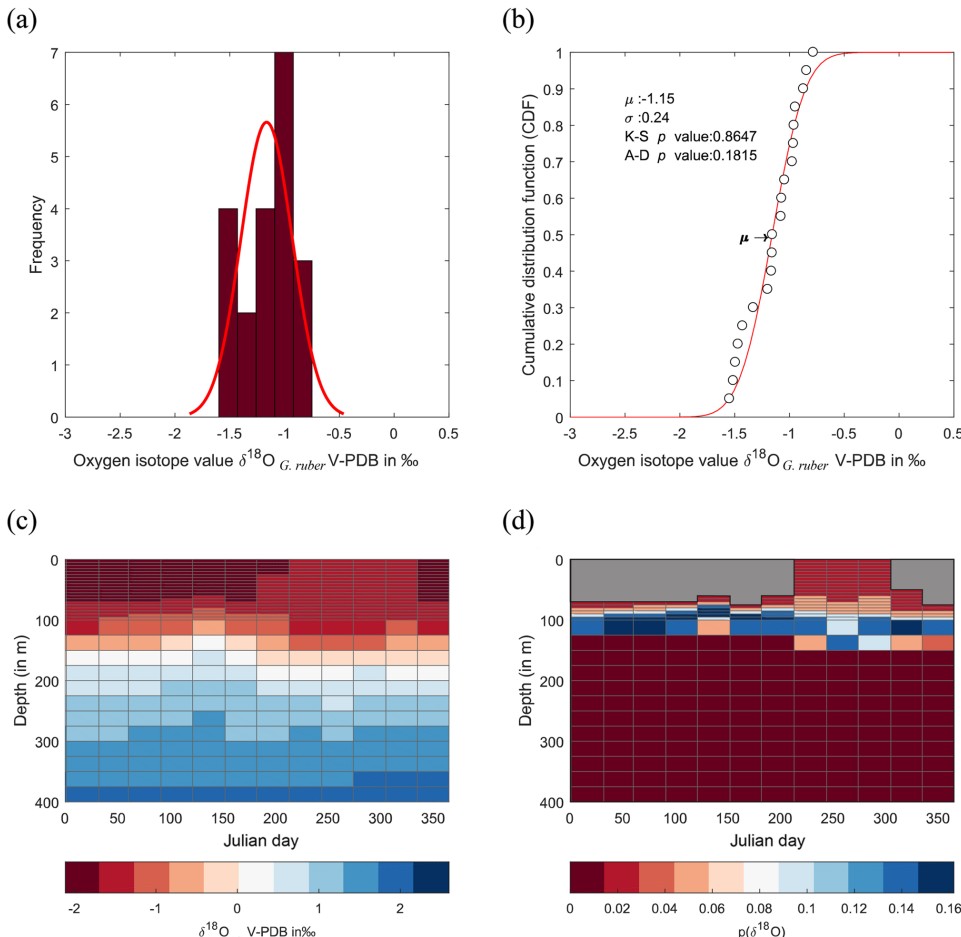

**Figure 8.** Calculated $\delta^{18}O$ probability ($p(\delta^{18}O)$). **(a)** Single-specimen isotope measurements for *G. ruber*, with a fitted normal distribution. **(b)** These data are used to produce a cumulative distribution function plot (CDF), with statistical output for an Anderson–Darling test, and a Kolmogorov–Smirnov test following data normalization (failure to reject the null hypothesis at the 5 % confidence suggests the data are not statistically different from a standard normal distribution, red line in plot). **(c)** In situ $\delta^{18}O_{eq}$ values predicted from a regional-specific equation (LeGrande and Schmidt, 2006) and a rearranged form of (Kim and O'Neil, 1997) with WOA13 temperature and salinity values as input values. **(d)** Resultant calculated $p(\delta^{18}O)$; the probability of each discrete $\delta^{18}O$ is denoted as $p(\delta^{18}O)$ mapped upon the $\delta^{18}O_{eq}$ WOA13 values. The grey region represents the area between 0 m and the first probable depth that may have become overprinted during depth migration.

same work suggests that GAM comprises 32 % to 44 % of *T. sacculifer* shells. Determining how many of the $< F$ and $F$ chambers are altered by GAM is complicated because GAM calcite precipitates on the outer "exposed" edges/margin of the shell; the amount of GAM relates to the surface area. Now, by removing the final chamber, a section of this surface area would not have been exposed during GAM formation. Therefore, the size of the over-printing is a product of both the amount of GAM calcite and the surface area exposed. Our results should therefore be considered as the minimum deviation between $< F$ and $F$.

## 4.3 Covariance with size

The trends in size–isotope values have been grouped into what Berger et al. (1978) considered to be three types: "normal" showing enrichment with increased size; "reversed" showing depletion with increased size and "mixed" in which neither enrichment or depletion with increasing size occurs. From our data there is no statistical difference in the $\delta^{18}O$ of the three different size classes, despite the appearance of a "mixed" signal, meaning there is absence of evidence to state that the $\delta^{18}O$ of *T. sacculifer* is subjected to a size effect. Evidence from our final chamber comparison shows that individuals undergo depth migration. Berger et al. (1978) considered that such a scenario should result in a "normal"

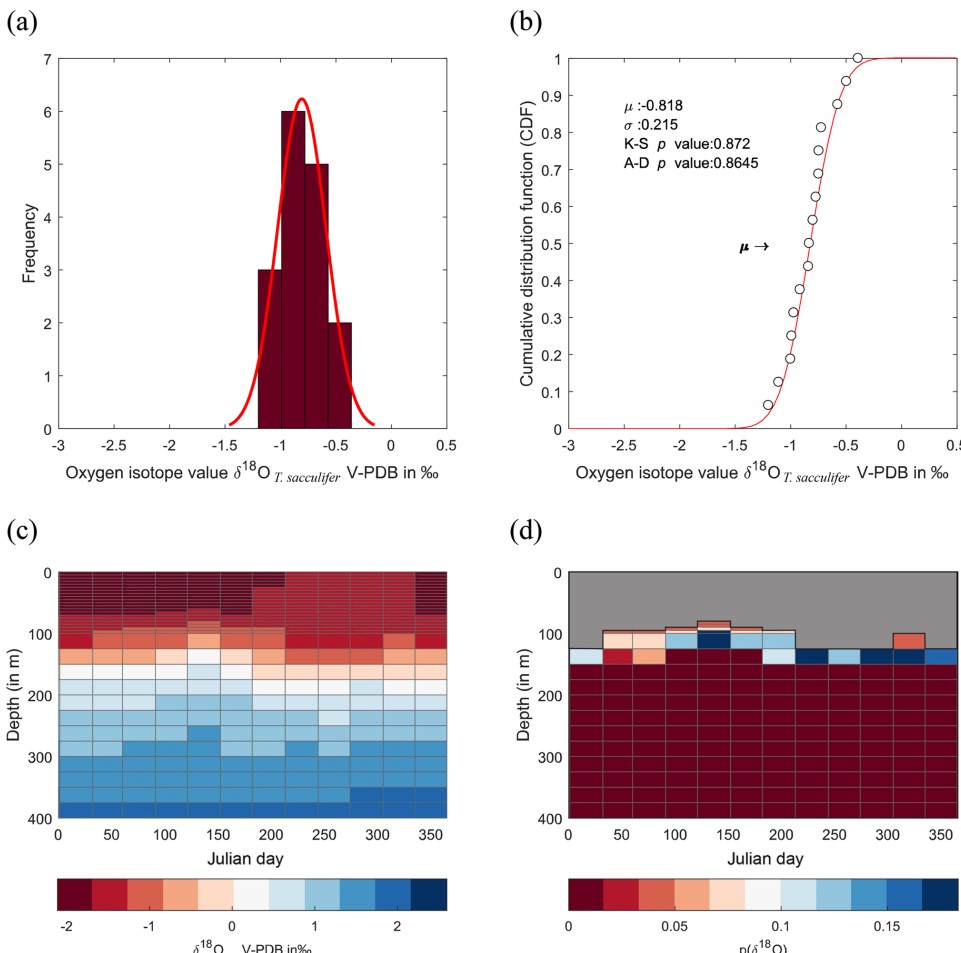

**Figure 9.** Calculated $\delta^{18}O$ probability ($p(\delta^{18}O)$). As per Fig. 8, with individual isotope values, CDF distribution, WOA13 $\delta^{18}O_{eq}$ and resultant $p(\delta^{18}O)$ but with *T. sacculifer*.

size–isotope trend; however, depth migration with size is not demonstrated in the $\delta^{18}O$ of the distinct size fractions. Berger et al. (1978) further considered that the "mixed" trend poses a problem in the interpretation of $\delta^{18}O$ solely in terms of depth migration. However, a study of population dynamics in the Red Sea indicate following reproduction at depth from the preceding generation, juveniles ascend in the water column to mature, where after these maturing foraminifera descend when reaching the reproductive size (Bijma and Hemleben, 1994; Hemleben and Bijma, 1994). How small forms would migrate from depth is unknown, although ascending particles due to low density do exist in the marine environment (Azetsu-Scott and Passow, 2004; Mari, 2008; Mari et al., 2007). This would lead the smallest shells to have calcite that was formed in deeper, colder waters, medium-sized shells to consist of calcite formed at the surface, in warmer waters and larger shells formed of calcite from deeper, colder waters. One caveat to such a scenario is that (Brummer et al., 1987, 1986) considered juvenile-neanic stages of the plank-

tonic foraminiferal life cycle to be less than 100–200 µm, distinctly smaller than the shell sizes measured here. Peeters et al. (1999) have shown that the size frequency distribution associated with the adult population of numerous planktonic foraminiferal species is distinctly gaussian in shape and thus variance around the mean should be considered as "dwarfs" and "giants" (Berger, 1971), thus a mixed signal may reflect extra-seasonal growth. A point of caution with size–isotope trends is that (Metcalfe et al., 2015) previously showed that such trends can either be consistent down core (e.g. *G. truncatulinoides*) or varying (e.g. *G. bulloides* and *G. inflata*) and therefore upscaling one relationship either spatially and/or temporally may lead to erroneous results.

### 4.4 Species-specific variability

The comparison between the $\delta^{18}O_{shell}$ values of the three species demonstrated that *G. ruber* ($\mu = -1.15\permil$) has a different mean $\delta^{18}O$ value and a different $\delta^{18}O$ distribution than either *T. sacculifer* ($\mu = -0.82\permil$) or *N. dutertrei* ($\mu =$

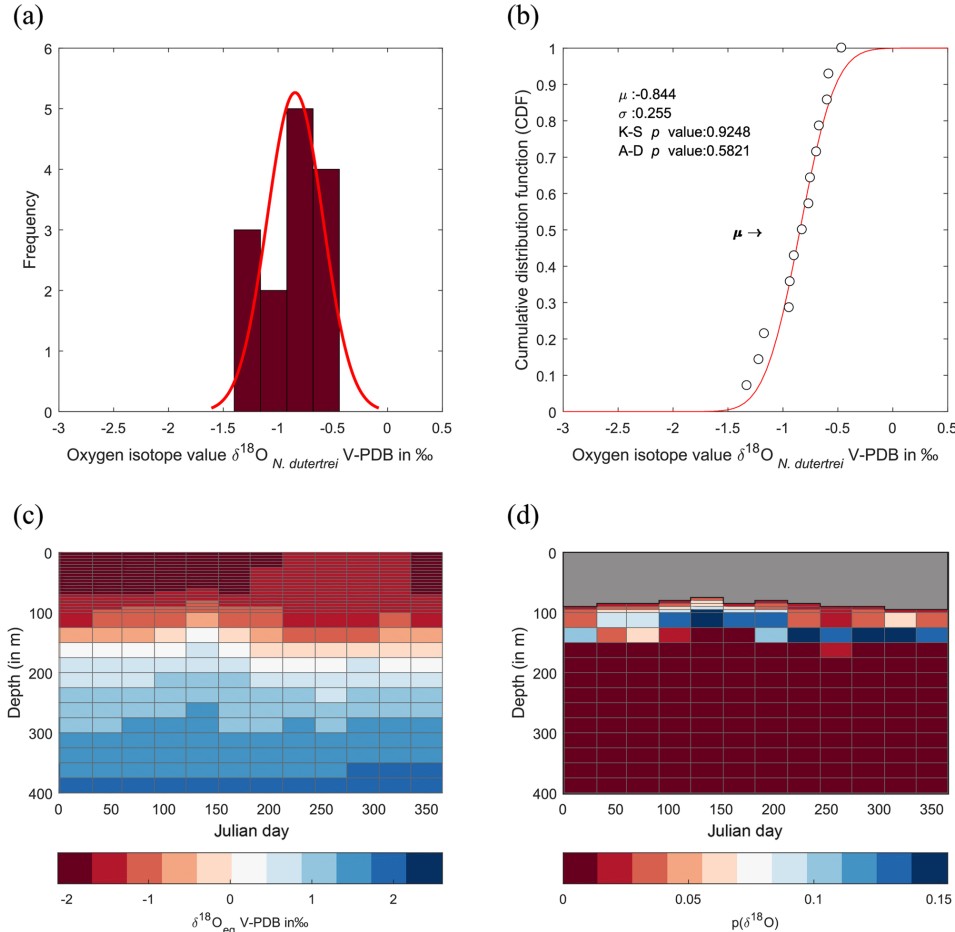

**Figure 10.** Calculated $\delta^{18}O$ probability ($p(\delta^{18}O)$). As per Fig. 8, with individual isotope values, CDF distribution, WOA13 $\delta^{18}O_{eq}$ and resultant $p(\delta^{18}O)$ but with *N. dutertrei*.

$-0.84\,‰$). Solving the palaeotemperature equations for each species using the mean $\delta^{18}O$ values gives an equivalent temperature of $24.1\,°C$ for *G. ruber*, $22.6\,°C$ for *N. dutertrei* and $22.5\,°C$ for *T. sacculifer*. This suggests a difference in depth habitat and/or season of growth between *G. ruber* and the other two species further highlighted by a comparison with the annual average and cumulative distribution functions (CDFs) for specific depths (Fig. 7b and c). Disentangling the signals of depth migration from seasonal habitat is complicated given the commonality between isotope values from similar depths and different seasons and vice versa. For instance, the same average isotope value will have a shallower depth habitat in May than in September. To illustrate this, at two specific depths (100 and 125 m; based on Fig. 6c), the $\delta^{18}O_{shell}$ of the foraminifera, corrected for the vital effect, was compared to the $\delta^{18}O_{eq}$ over the year for a number of discrete depth levels in the water column, to find out at which depth level(s) a given species could grow, assuming a uniform shell flux over the year (Fig. 7a). For *N. dutertrei* and *T. sacculifer* these potential depths and seasons of growth

are similar, following from the fact that their mean $\delta^{18}O_{shell}$ and $\delta^{18}O_{shell}$ variability is not significantly different. It was found that *T. sacculifer* and *N. dutertrei* could potentially occur year-round at $\sim 125$ and at $\sim 100\,m$ from respectively August to December and February to June. *G. ruber* in its place reflects the year-round temperature at $100\,m$ of depths and autumn/winter temperatures (August to December) at a depth of $125\,m$. Wit et al. (2010) stated that the variability within single-shell $\delta^{18}O$ measurements could be a proxy for seasonality (Ganssen et al., 2011; Vetter et al., 2017), which was inferred from measurements of single species (*G. ruber*) for multiple core locations to test this inference. Here we tested whether different species are influenced by seasonality in a similar or dissimilar way.

Our results imply that species can be used as indicators of year-round seasonality, because the variability in single-shell $\delta^{18}O$ matches the variability in annual temperature derived from the climatological average of WOA13, but only at species-specific depths (*T. sacculifer* and *N. dutertrei* for $125\,m$, and *G. ruber* for $100\,m$). The probability plots of the

season–depth habitat, as indicated by Figs. 8–10, show that the calcification depth recorded by the shell $\delta^{18}O$ is a narrow interval between 50 and 200 m. Despite evidence to the contrary, $\delta^{18}O$ does not implicitly record sea surface temperatures, collection of foraminifera by SCUBA (Bird et al., 2018; Spero, 1998) and net collection (Ottens, 1992; Kroon and Ganssen, 1989; Ganssen and Kroon, 1991) at or in close proximity to the sea surface represents a part of, but not their full, life cycle. This situation is further exacerbated by both a shallow or deep mixed layer giving a potential homogeneous $\delta^{18}O$ signal from surface to deep (Fig. 1) and the unknown quantity of the vital effect when attempting to derive depths from core top material. It is worth reiterating, here, several conclusions of previous studies (Wilke et al., 2006). Foraminiferal depth habitat is a continuous variable from zygote fusion to eventual reproduction-induced mortality. However, chambers represent a distinct event covering a short period of time ($\sim 12$ h); the calcification depths of chambers therefore reflect discrete intervals along this continuous depth habitat. As chamber size increases progressively, in normal forms (Berger, 1969), from the earliest to the final chamber the contribution of each chamber to the cumulative signal increases iteratively and can be approximated by a mass balance (e.g. Wilke et al., 2006). As the shell sinks through the water column, during its life, the signal will become progressively skewed toward a deeper "colder" signal. Modification of this signal via crust formation or GAM calcite will bias the signal further toward higher $\delta^{18}O$ and a colder signal. The depth habitat of foraminifera is not static globally; instead, its dynamism represents a complex interaction between food, temperature, water column structure and, where appropriate, light. Discrepancies between previously published work should not be considered in depth but on the various attributes of the water column present, as it is those parameters altering with depth that ultimately allow foraminiferal growth to occur.

## 5 Conclusions

To gain more insight into biological and ecological processes that influence the $\delta^{18}O_{shell}$ of planktonic foraminifera, three research questions with associated hypotheses were tested. First, we tested depth migration and found that a significant difference in $\delta^{18}O$ between the final chamber ($\delta^{18}O_F$, $\mu = -1.23\,‰$) and the test minus the final chamber is observed in *T. sacculifer*. This difference in $\delta^{18}O_{shell}$ is equal to a temperature difference of 1 °C, suggestive that the final chamber is formed via depth migration in waters that are approximately 1 °C cooler than the chambers formed prior. Second, we tested covariance with size and found that despite evidence for depth migration during the life and growth of *T. sacculifer* there is an absence for a size effect on *T. sacculifer* with no statistical difference in the $\delta^{18}O_{shell}$ of the three different size classes. Third, we tested species-specific

$\delta^{18}O$ variability to quantify the effect upon the populations from proxy archives. Comparison between *T. sacculifer* ($\mu = -0.82\,‰$), *G. ruber*$_{white}$ ($\mu = -1.15\,‰$) and *N. dutertrei* ($\mu = -0.84\,‰$) indicate that *G. ruber* has both a significantly different mean and variability in $\delta^{18}O$, suggestive that the species lives in warmer shallower waters (i.e. $\sim 90$–120 m vs. $\sim 100$–130 m). However, inferences about depth and/or seasonal habitat is complicated by the fact that similar $\delta^{18}O_{eq}$ values occur in both time and depth. It is possible, based upon our results that *T. sacculifer* and *N. dutertrei* could potentially occur year-round at 125 m of depth and at 100 m of depth from respectively January to August and February to June. *G. ruber* in its place, reflects the year-round temperature at 100 m of depths, and autumn/winter temperatures (August to December) at a depth of 125 m. These results highlight the complicated nature of interpreting oxygen isotopes even for the modern record in line with previous findings (Kretschmer et al., 2018; Roche et al., 2018). Depth migration, size and species-specific variability all influence the values of $\delta^{18}O$ within a foraminiferal shell and therefore the resultant palaeoclimate reconstructions conclusions drawn from their isotope values.

*Data availability.* TS9

*Author contributions.* FJCP and BM designed the study and HP performed data collection and analysis under supervision of BM and FJCP, upon material collected by FJCP. HP and FJCP performed statistical analysis. BM produced the figures. All the authors contributed to the writing of the manuscript.

*Competing interests.* The authors declare that they have no conflict of interest.

*Acknowledgements.* This work is a contribution to the RETRO project, a European Science Foundation (ESF)/EUROMARC project, jointly funded by research councils of the Netherlands (NWO), Norway (RCN), France (CNRS/INSU) and Germany (DFG). The captain and crew of R/V *G. O. Sars* and Chief Scientist Trond Dokken are thanked for the collection and sharing of research material. FP and BM wish to acknowledge sponsoring from the Netherlands Organisation for Scientific Research (NWO) open round project grant number NWO/822.01.0.19. The data and work presented here reflect work produced for the fulfilment of a VUA Earth Surface Processes, Climate and Records (ESPCaR) MSc Research Project (Hilde Pracht). SEM images in Fig. 2 produced by Saskia Kars (VUA) and Frank J. C. Peeters.

Edited by: Lennart de Nooijer
Reviewed by: Jody Wycech and Takashi Toyofuku

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

**Remarks from the typesetter**

**TS9**  Please provide a statement on how your underlying research data can be accessed. If the data are not publicly accessible, a detailed explanation of why this is the case is required. The best way to provide access to data is by depositing them (as well as related metadata) in reliable public data repositories, assigning digital object identifiers (DOIs), and properly citing data sets as individual contributions. Please indicate if different data sets are deposited in different repositories or if data from a third party were used. If no DOI is available, assets can be linked through persistent URLs to the data set itself (not to the repositories' home page). This is not seen as best practice and the persistence of the URL must be secured.