# Peer review of "Oxygen isotope composition of final chamber of planktic foraminifera provides evidence for vertical migration and depth integrated growth"

_Biogeosciences, 2018_

## Referee Comment (RC1) · Ph.D. Wycech (Referee) · 8 May 2018

General Comments

This is an interesting paper that thoroughly investigates the $\delta18O$ signals recorded by planktic foraminifera using novel small sample isotope ratio measurements. The objectives of the paper are clearly stated and quantitatively investigated with statistical hypothesis testing. The application of rigorous statistical techniques including the clear statement of the null hypothesis and explanation of each test with associated assumptions is excellent and should be utilized more in the discipline. Foraminifera are broadly studied by ecologists, oceanographers, and paleoclimatologists, which makes this paper well-suited for publication in Biogeosciences. I fully encourage the publication of this paper after the authors consider some minor suggestions that will provide clarification and more context for their results.

Specific Comments

I recommend the authors also cite Kozdon et al. (2009). Kozdon et al. (2009) used SIMS to measure the d18O values of the ontogenetic calcite and reproductive crust of N. pachyderma to reconstruct the species' water depth migration.

Kozdon, R., Ushikubo, T., Kita, N.T., Spicuzza, M.J., and Valley, J.W., 2009, Intratest oxygen isotope variability in the planktonic foraminifer N. pachyderma: Real vs. apparent vital effects by ion microprobe: Chemical Geology, v. 258, no. 3-4, p. 327–337, doi: 10.1016/j.chemgeo.2008.10.032.

Page 5 - Lines 20-22: How many G. ruber and N dutertrei were analyzed or were they just not weighed and measured for size? Perhaps breaking the description of T. Sacculifer analyses into a subsection within 2.1 will help (or changing the title of this section to be specific to T. sacculifer). What were the additional steps used for T. sacculifers prior to stable isotope analysis? If the additional steps involve the heated block, the authors should also describe how the d18O analyses were performed for the G. ruber and N. dutertrei shells.

Page 6 - Lines 20-21: What is the vital effect d18O correction? I think readers will be able to better interpret Fig. 6b with clarification. Is the correction necessary if you use the d18O-temperature calibration of Mulitza et al. (2003) instead of Kim and O'Neil (1997) (see comment regarding Page 6 – Line 30)?

Page 6 - Line 30: Why use Kim and O'Neil (1997) instead of a foram-based calibration such as Mulitza et al. (2003)? I suggest adding a sentence or two for explanation.

Page 8 - Lines 7-9: I'm not sure how Figs. 5c-d show statistical significance (although the trend is quite obvious visually). Perhaps you could plot the 95% confidence interval on the robust regression (iteratively reweighted least squares) to show the slope is statistically different from zero. Alternatively, you could leave the plot as is and change the sentence to state D18O is dependent upon the d18O of the measured fragment.

Page 9 – Lines 18-19: "above and below 100 m" is a bit vague. Note which water depths have temperatures of 24.5°C and 25.5°C at this site. If 25.5 °C measured from the <F fragment is not in the upper 0-50 m, that would suggest the fragment is also partly composed of GAM crust (see comment for Page 10 -Lines 18-21).

Page 10 – Lines 18-21: I've used image processing of shell walls in cross-section, which suggests that GAM crust composes 32-44% of T. sacculifer shells. I've also used SIMS to analyze the $\delta$18O of PREGAM and GAM calcites in the penultimate chamber of Holocene and Pliocene T. sacculifer shells from several Atlantic and Pacific sites, and have found that the GAM is $\sim$1‰ higher in $\delta$18O than the PREGAM. That work is still in prep, but you could cite one of the following abstracts or my dissertation (refs below). I think Lines 18-21 undervalue the issues with GAM crust so at the very least, I suggest noting that the presence of GAM crust in the <F fraction may skew your results towards higher $\delta$18O (colder temperatures and a deeper depth habitat).

Wycech, J (2017) Novel techniques and approaches to enhance the fidelity of foraminiferal paleoclimate records: University of Wisconsin-Madison, 231 p.

Wycech J, Kelly DC, Kozdon R and Valley J (2016) On the fidelity of $\delta$18Osw reconstructions: comparing paired foraminiferal Mg/Ca-$\delta$18O values from conventional and in situ techniques. Poster, International Conference on Paleoceanography, P-287.

Wycech J, Kelly D, Kozdon R, Fournelle J and Valley JW (2013) Warm Tropical Sea Surface Temperatures During the Pliocene: A New Record from Mg/Ca and $\delta$18O In Situ Techniques. Poster, American Geophysics Union Fall Meeting, PP53C-2016.

Page 11 – Lines 32-33: This section really highlights issues with the $\delta$18O proxy, and suggests that the values do not reflect sea-surface conditions. I love the statistical methods used to reconstruct depth habitats, but the estimates seem too deep relative to culturing and plankton tow observations. For example, foram culturing scientists collect wild G. ruber and T. sacculifer from the upper 20 m of the water column so the probability of finding these species in the upper 50 m is not zero as your $\delta$18O results suggest. I recommend shifting the focus of the section away from where the forams are actually calcifying and focus instead on your $\delta$18O results, why they deviate from field observations, and what this means for others who try to use $\delta$18O to infer ocean conditions.

Page 19 (Fig. 2): I suggest adding labels in the figure to identify the s.l. and s.s. morphotypes of G. ruber (or add a note into the figure caption which numbers are s.l. and s.s.).

Page 21 (Fig. 4B): I think it may help to color code the points based on the morphologies noted in figure 2. I like the inset images and I think you should keep them, but I was curious about the morphology of the shells with high whole shell area and high final chamber area that didn't have a corresponding inset photo (points in the upper right).

Technical Corrections

Even though the mass spec produces $\delta$18O values to many decimal places, values beyond the tenth place are uncertain so d18O values should only be reported to the tenth decimal place (i.e., 0.1‰ precision).

Page 1 - Lines 16-17: I suggest noting the direction of this difference, I.e. "We show that the d18O of the final chamber ($\delta$18OF) is 0.2 ‰ $\pm$ 0.4 ‰ ($1\sigma$) higher than the d18O value of the test minus the final chamber ($\delta$18O<F) of T. sacculifer"

Page 1 - Line 17: Specify if sigma is standard deviation or standard error. Also, note

how many shells you analyzed in the parentheses "(n=__)"

Page 2 - Line 16: Remove the double parentheses

Page 3 – Line 2: The word "do" is not necessary

Page 4 - Line 10: Change "were" to "was" if only one test was performed (ANOVA with a post-hoc test)

Page 4 – Line 16: I stumbled over the statement "[. . .] our third objective Seasonality is a [. . .]" Page 5 – Line 2: Condense the sentence to read "[. . .] same location, which would mean [. . .]"

Page 8 - Line 16: I suggest separating these sentences to read something like "[. . .](Figure 6b). An ANOVA to test whether the species had equal means resulted in a p value of 0.0001 and led to a rejection [. . .]"

Page 10 - Line 8: Change negative to positive (colder SSTs = more positive foram d18O values)

Page 10 - Line 43: Use parentheses only around the year, i.e. "Berger et al. (1978b)"

Page 10 - Line 28: Change the comma to a period

Page 11 - Line 5: Use parentheses only around the year, i.e. "Brummer et al. (1987, 1986)"

Page 11 - Line 6: Use parentheses only around the year, i.e. "Peeters et al. (1999)"

Page 11 - Line 18: Add parentheses around the figure reference

Page 11 - Line 27-29: Use parentheses only around the year, i.e. "Wit et al. (2010)". The verbiage is a bit awkward. I suggest dividing it into two sentences, I.e. "Wit et al. (2010) stated [. . .], which was inferred from measurements of single species (G. Ruber) at multiple core locations. Here we test [. . .]".

Page 12 - Line 3: The verbiage is a bit awkward. I suggest, "First, we tested depth

migration and found [. . .]".

Page 12 - Line 7: Similar to line 3 I suggest, "Second, we tested covariance with size and found [. . .]"

Page 12 - Line 8-9: "found" is used twice in the sentence. I suggest deleting the second one, "[. . .] the three measured size classes."

Page 12 – Line 9: If you use my suggestion for lines 3 and 7, this should be consistent, i.e. "Third, we tested [. . .]."

Page 12 – Line 10: Divide into two sentences. "[. . .] archives. Comparison between [. . .]"

Page 12 – Line 21: Remove period after "BM"

Page 14 – Line 12: Remove the extra ", "

Page 17 – Line 24: Remove the extra ", "

Page 17 – Line 31: Add superscript to 18 in $\delta$18O

Page 24 – Line 8: italicize the latin "in situ"

Page 24 (Fig. 6 caption): Add in what the whiskers represent (e.g. 95% confidence interval) and if the horizontal lines within the boxes are the median or mean. The lines are typically medians, but the text compares means of the datasets so you may want to show both the mean and the median (perhaps one as a bold line and one as a dashed line?)

Page 26, 27, 28 (Fig captions): Note what "p($\delta$18O)" is. I initially thought it was a p-value.

---

## Editor Comment (EC1) · L.J. de Nooijer (Editor) · 19 Jun 2018

Dear Dr Pracht and co-workers,

we are awaiting the second reviewer's comments. But they are under way: please have some more patience.

Sincerely,

Lennart de Nooijer

---

## Referee Comment (RC2) · T. Toyofuku (Referee) · 27 Jun 2018

This is a nice research that clearly verified their own constructed hypotheses under excellent research philosophy by accumulated precise data. The illustrations are also very beautiful and meaningful. The study should be published on "Biogeosciences". The broader audience of the journal will enjoy to read this article.

I have just small question. For Question 2 of P10L25, can you expect that the difference among size groups will become statistically significant as the number of samples

increases?

It is quite happy for me if the comments are useful for authors what is indicated below.

P6L8 Put "2012" after Ishimura

P11L6 (Peeters et al., 1999) → Peeters et al. (1999)

P13L7 Be→Bé. Check other description about him through the manuscript. The both descriptions are found.

P13L11 Check the authors' name→Bé, A. W. H., Van Donk, J., Hecht, A. D., and Savin, S. M.: Oxygen-18 Studies of Recent Planktonic Foraminifera, Science, 173, 167-169, 1971.

P13L13 Put "a" after foraminifer Be, A. W. H., Spero, H. J., and Anderson, O. R.: Effects of symbiont elimination and reinfection on the life processes of the planktonic foraminifer"a" Globigerinodies sacculifer , Marine Biology, 70, 73-86, 1982.

P13L18-21 Same study is referred twice.

P13L22 Remove "-" from journal title: Bijma, J. and Hemleben, C.: Population dynamics of the planktic foraminifer Globigerinoides sacculifer (Brady) from the central Red Sea, Deep"-"Sea Research I, 41, 485-510, 1994.

P13L24 Remove "a" from planktonic foraminifer"a"

P13L30 Globigerinidae is not italicized.

P14L1 Put Journal name, volume and pages. Brummer, G.-J. A., Hemleben, C., and Spindler, M.: Planktonic foraminiferal ontogeny and new perspectives for micropalaeontology, Nature, 319, 50, 1986.

P14L5 Caron et al. is published in 1982.

P14L10 Put page numbers for Davila and Hunt as 117-145.

P14L11 Change to lower-case letters with the each word other than Oxygen. Remove additional "," before "Science".

P14L17 Faber et al. is cited twice.

P14L24 "Globorotalia truncatulinoides" would be italicized.

P14L27 Pages would be 1337-"1349".

P14L31 Italisize "Orbulina universa"

P14L32 The page numbers would be "145-166".

P15L1 The volume and page numbers would be "30:141-171" for Hemleben and Spindler (1983).

P15L24 Check the usage of abbreviations about "Clim. Past Discuss". Other references are shown in full spells.

P15L27 Page numbers can be shown as "413-427".

P16L1 "1023–1031" will be shown. Check about double "2018".

P16L10 Check the title of Mix (1987). Isn't it "glaciation" instead of "deglaciation"? Show page numbers if it is possible.

P16L18 Show volume and page numbers of Peeters et al (2002). Maybe 34, 269-291. Remove double "2002".

P17L6 Check about double ",".

P17L7 Put "The" at the beginning of journal title.

P17L28 Remove "2017" before page numbers.

P17L30 Remove "n/a-n/a,". "18" would be superscript.

P17L31 "18" would be superscript.

---

## Author Comment (AC1) · 17 Jul 2018

We have noticed that panel B in Figure 6, the box plot of the different species, is incorrect as presented in the manuscript. This does not affect either the text or statistics which use the correct data, in constuction of the box plot the box seem to have been reversed and therefore has an incorrect vital effect. Here we present a correct version of the figure.

[Figure]

[Figure]

**Fig. 1.** Original figure 6

[Figure]

**Fig. 2.** Corrected figure 6

---

## Author Response (AR1)

Response to Reviewer 2. Takashi Toyofuku

Outlined below is our responses to their questions:

We thank the reviewer for their comments and thorough reviewing and will enact their textual suggestions. We do note however that at P14L17 Faber is not cited twice, it's a part 1 and part 2 paper.

[Figure]

Question 2 of P10L25, can you expect that the difference among size groups will become statistically significant as the number of samples increases?

Its an interesting question, the simplest answer is that we don't know whether by adding more data our results/significance would change. That being said, we do wonder if we expanded the size range to larger and smaller size classes whether we would see a different or a continuation of the trend. Or if we measured more specimens we would get a different result, essentially we can only know what we have measured with our current N and method. We know that, from a previous study, by increasing the number of size fractions or pooled analysis globally there seems to be a size isotope relationship (Ezard et al., 2015). Whether that holds true for individual sites or is the 'global' signal is intriguing.

Reference

Ezard, T.H.G., Edgar, K.M., and Hull, P., 2015. Environmental and biological controls on size‐specific $\delta$13C and $\delta$18O in recent planktonic foraminifera, Paleoceanography, doi: 10.1002/2014PA002735

[Figure]

Biogeosciences Discuss.,
https://doi.org/10.5194/bg-2018-146-AC3, 2018

[Figure]

We thank the reviewer for their feedback and interesting comments, and apologize for not being able to have an online discussion regarding their points. Outlined below is our responses to their questions:

I recommend the authors also cite Kozdon et al. (2009). Kozdon et al. (2009) used SIMS to measure the d18O values of the ontogenetic calcite and reproductive crust of

N. pachyderma to reconstruct the species' water depth migration.

We shall include the paper at an appropriate spot.

Page 5 - Lines 20-22: How many G. ruber and N dutertrei were analyzed or were they just not weighed and measured for size? Perhaps breaking the description of T. Sacculifer analyses into a subsection within 2.1 will help (or changing the title of this section to be specific to T. sacculifer).

We shall add in an additional section as follows: "2.X Specific methodology for Question 3 To determine whether species of planktonic foraminifera from the same geographic location share the same or similar single specimen d18O(shell) variability specimens of G ruber (n = X) and N. dutertrei (n = X) were picked from the same interval. These shells underwent the same methodology outlined in section 2.1 for photographing, weighing and isotope analysis."

What were the additional steps used for T. sacculifers prior to stable isotope analysis? If the additional steps involve the heated block, the authors should also describe how the d18O analyses were performed for the G. ruber and N. dutertrei shells.

No all isotope values are based upon a heated block, the additional steps performed on T. sacculifer are outlined in sections 2.2 and 2.3. We shall add in the following text in line 22 (Pg. 5) so that it reads: "underwent additional steps, outlined in section 2.2 (dissection of chambers) and section 2.3 (size fraction), prior to stable isotope analysis."

Page 6 - Lines 20-21: What is the vital effect d18O correction? I think readers will be able to better interpret Fig. 6b with clarification. Is the correction necessary if you use the d18O-temperature calibration of Mulitza et al. (2003) instead of Kim and O'Neil (1997) (see comment regarding Page 6 – Line 30)?

We strictly use the equation by Kim and O'Neil as the definition for the d18Oeq, similar to previous studies by our and many other groups. The use of one equation to define (inorganic) equilibrium is preferred in order to avoid different equations for different

species or even for different shell sizes. Plankton pump and multinet collected specimens from the surface mixed layer were used to establish the value for the vital effect. For both species G. ruber and G. trilobus we used a d18Ove correction of -0.48 per mille. No vital effect correction was used for the non spinose species Neogloboquadrina dutertrei. These data are from Peeters et al., 2004 (Nature) and from Peeters et al., 2000 Ph.D. thesis). The error on the estimation of the vital effect is 0.15 per mille (1 s.d.).

Here we correct the measured values for vital effects of the species and perform statistical tests on both the corrected and uncorrected values. We will add in the following text: "In order to account for similar absolute measured values between species which are not produced by concurrent depth or seasonal preferences between species but instead by species-specific disequilibria from values obtained from ambient seawater equilibrium (so called 'vital effects') a correction was applied. The d18O shells were corrected by -0.48 per mil for G. ruber and G. sacculifer. "

Page 6 - Line 30: Why use Kim and O'Neil (1997) instead of a foram-based calibration such as Mulitza et al. (2003)? I suggest adding a sentence or two for explanation

See above: we want to avoid different equations and define strictly use K&O with an extra term for the vital effect. As such the temperature equations for different species may be offset from one another but all equations are parallel having a similar slope at a given temperature. We agree with the reviewer that a foraminiferal based calibration may be more suited in some instances, however these calibrations focus solely upon temperature as a driving factor for d18O (apart from the culture based ones). Carbonate ion is known to influence the d18O, and a number of these calibrations do not take this into account. Kim and O'Neil (1997) is used because it represents unmodified equilibrium, and whilst it also doesn't take into account the CIE it is a useful base to build upon as it is not subjective. However, it is easy to vary the input using a number of published calibrations to test the sensitivity of these results.

Page 8 - Lines 7-9: I'm not sure how Figs. 5c-d show statistical significance (although the trend is quite obvious visually). Perhaps you could plot the 95% confidence interval on the robust regression (iteratively reweighted least squares) to show the slope is statistically different from zero. Alternatively, you could leave the plot as is and change the sentence to state D18O is dependent upon the d18O of the measured fragment.

We tested the significance (r being not zero) using Pearson's correlation coefficient: The critical value for the absolute value of the correlation coefficient for alpha = 0.05 and N=57 (Deg. of freedom = 55) is 0.273 for DF=50 and 0.250 for DF = 60 (our table does not give a critical value for DF=55). Since our correlations coefficients are higher than these Critical values we may conclude they are different from zero. The correlation coefficients are also significant at the alpha = 0.01 level for which the highest critical values is 0.354 (for DF = 50). The graphs indicate that there is a relationship between the d18O value of the shell minus the final chamber and the d18O difference between the final chamber and the remaining shell value. This can be interpreted that the difference between final chamber and remaining shell may be a function of the surface water temperature as the (temperature difference between final chamber and shell minus final chamber) decreases with decreasing temperature. Potentially for low SST's we may face a situation that the final chamber may even be warmer compared to the remaining shell value.

Page 9 – Lines 18-19: "above and below 100 m" is a bit vague. Note which water depths have temperatures of 24.5âŮęC and 25.5âŮęC at this site. If 25.5âŮęC measured from the <F fragment is not in the upper 0-50 m, that would suggest the fragment is also partly composed of GAM crust (see comment for Page 10 -Lines 18-21).

We chose to be a bit vague with the depths, however, we will add a section to the text considering the potential for the two proportions to have GAM addition. We will further consider that if GAM calcite precipitates upon the outside of the shell, then it could be that the proportion of GAM for F and <F may have different amounts if F has a larger surface area.

Page 10 – Lines 18-21: I've used image processing of shell walls in cross-section, which suggests that GAM crust composes 32-44% of T. sacculifer shells. I've also used SIMS to analyze the $\delta$18O of PREGAM and GAM calcites in the penultimate chamber of Holocene and Pliocene T. sacculifer shells from several Atlantic and Pacific sites, and have found that the GAM is âĹij1‰ higher in $\delta$18O than the PREGAM. That work is still in prep, but you could cite one of the following abstracts or my dissertation (refs below). I think Lines 18-21 undervalue the issues with GAM crust so at the very least, I suggest noting that the presence of GAM crust in the <F fraction may skew your results towards higher $\delta$18O (colder temperatures and a deeper depth habitat).

In shells of living O. universa we have noted (after cracking shells and observing under SEM) that there is a linear relationship between the width of the inner wall and the width of outer wall (as delimited by the POM) in 'surface'-'mixed layer' samples that is not seen in the settling flux/dead population. We considered whether this reflects potential gam-calcite, therefore we look forward to seeing your results published and will include a citation.

This could explain why sometimes the last chamber is warmer compared to the remaining shell. However, if gam-calcite forms on the outer 'exposed' edges on the outer margin of the shell (and on the outer chambers) then the amount of GAM calcite would relate to the surface area exposed and how much GAM calcite may be added to the previous chambers (whole shell minus the last one) and the last chamber. Attached is a x-ray through a shell showing the complicated 'thickening'. We shall expand our discussion to incorporate this.

Page 11 – Lines 32-33: This section really highlights issues with the $\delta$18O proxy, and suggests that the values do not reflect sea-surface conditions. I love the statistical methods used to reconstruct depth habitats, but the estimates seem too deep relative to culturing and plankton tow observations. For example, foram culturing scientists collect wild G. ruber and T. sacculifer from the upper 20 m of the water column so the probability of finding these species in the upper 50 m is not zero as your $\delta$18O results

suggest. I recommend shifting the focus of the section away from where the forams are actually calcifying and focus instead on your $\delta$18O results, why they deviate from field observations, and what this means for others who try to use $\delta$18O to infer ocean conditions.

Foraminiferal d18O does not reflect SST for two reasons, the first is that the vital effect leads to an offset and thus we must either alter the equilibrium line of KO or the individual d18O of the foraminifera. Secondly reflects the depth habitat of foraminifera: The depth habitat of foraminifera is a continuous variable however the calcification depths ('apparent calcification depths) represent a series of discrete intervals within the total depth habitat of a single specimen. Foraminifera construct a shell in which along the whorl the chamber size is 'exponentially' increasing in size, so that the cumulative fraction of each chamber to the total shell increases iteratively. Therefore, whilst foraminifera can be caught (via the authors own experience and as stated by the reviewer) in tows or via divers in the upper water column the seafloor shells are themselves skewed toward either a 'colder signal' or the signal with the greatest 'mass'. This mass balance approach can be seen, and is outlined, in Wilke et al., 2006. We completely agree with the reviewer that foraminifera do not catagorically record sea-surface conditions because of this 'weighted averaging'/cumulative mass balance.

I agree with the reviewer that "the probability of finding these species in the upper 50 m is not zero as your $\delta$18O results suggest" and will alter both the text and figure accordingly, instead what these plots show is the depth of the apparent weighted average signal. One should interpret the zero probability above the interval with 'zero' probability as being part of the depth habitat, to make this more clear, we will mask the upper section to indicate that this should be seen as part of the depth habitat.

This brings us back to the reviewers comment regarding species-specific d18O equations, the culture derived d18O-temperature approximations took pains to remove the field-grown portion of the shell (via dissection and subsequent pooling of culture-grown chambers), with respect to other field based methods (such as from tows or pump

samples) this mass balance might skew the results.

ref: Wilke, I., Bickert, T., and Peeters, F.J.C., 2006. The influence of seawater carbonate ion concentration [CO32−] on the stable carbon isotope composition of the planktic foraminifera species Globorotalia inflata, Marine Micropaleontology,

Page 19 (Fig. 2): I suggest adding labels in the figure to identify the s.l. and s.s. morphotypes of G. ruber (or add a note into the figure caption which numbers are s.l. and s.s.).

Will be added in a revised version of the text

Page 21 (Fig. 4B): I think it may help to color code the points based on the morphologies noted in figure 2. I like the inset images and I think you should keep them, but I was curious about the morphology of the shells with high whole shell area and high final chamber area that didn't have a corresponding inset photo (points in the upper right)

We will endeavor to color code the points based on the morphologies noted in figure 2, but this might be a subjective approach (i.e. dependent upon the interpreter), we will see if another morphological parameter (such as deviation from a circle) could be used instead. However, we shall definitely add the morphology of the upper right.

Technical Corrections Even though the mass spec produces $\delta18O$ values to many decimal places, values beyond the tenth place are uncertain so d18O values should only be reported to the tenth decimal place (i.e., 0.1‰ precision).

Whilst, the reviewer is correct regarding isotope values there is also the problem of rounding error. We will add within the text a statement to that affect: "Isotope values are reported to 2 decimal places, however this should not be misconstrued as reflecting a degree of certainty but to prevent rounding error."

Page 1 - Lines 16-17: I suggest noting the direction of this difference, I.e. "We show that the d18O of the final chamber ( $\delta18OF$) is 0.2 ‰\$±0.4 ‰ (1$\sigma$) higher than the

d18O value of the test minus the final chamber ( $\delta18O<F$) of T. sacculifer"

We will modify the text accordingly

Page 1 - Line 17: Specify if sigma is standard deviation or standard error. Also, note how many shells you analyzed in the parentheses "(n=__)"

Sigma by itself is standard deviation, the symbol for standard error is sigma subscript SE or x-hat We will modify the text accordingly, by adding the n

Page 2 - Line 16: Remove the double parentheses

We will modify the text accordingly

Page 3 – Line 2: The word "do" is not necessary

We will modify the text accordingly

Page 4 - Line 10: Change "were" to "was" if only one test was performed (ANOVA with a post-hoc test)

A single test was performed, we will modify the text accordingly

Page 4 – Line 16: I stumbled over the statement "[ ...] our third objective Seasonality is a [...]"

We agree and have modified the text as follows:

"Having focused upon a single species for the first two objectives, our third objective focuses upon the variability of foraminifera isotope values, which are considered to represent seasonality, and whether fossil shells from different species have similar d18Oshell variability. "

Page 5 – Line 2: Condense the sentence to read "[...] same location, which would mean [...]"

We will modify the text accordingly

Page 8 - Line 16: I suggest separating these sentences to read something like "[...] Figure 6b). An ANOVA to test whether the species had equal means resulted in a p value of 0.0001 and led to a rejection [...]"

We will modify the text accordingly

Page 10 - Line 8: Change negative to positive (colder SSTs = more positive foram d18O values)

We will modify the text accordingly

Page 11 - Line 18: Add parentheses around the figure reference

We will modify the text accordingly

Page 10 - Line 28: Change the comma to a period

We will modify the text accordingly

Page 11 - Line 27-29: The verbiage is a bit awkward. I suggest dividing it into two sentences, I.e. "Wit et al. (2010) stated [...], which was inferred from measurements of single species (G. Ruber) at multiple core locations. Here we test [...]".

We agree, we will modify the text

Page 12 - Line 3: The verbiage is a bit awkward. I suggest, "First, we tested depth migration and found [..]". Page 12 - Line 7: Similar to line 3 I suggest, "Second, we tested covariance with size and found [..]". Page 12 – Line 9: If you use my suggestion for lines 3 and 7, this should be consistent, i.e. "Third, we tested [...]."

We will modify the text accordingly

Page 12 - Line 8-9: "found" is used twice in the sentence. I suggest deleting the second one, "[...] the three measured size classes."

We will modify the text accordingly

Page 12 – Line 10: Divide into two sentences. "[...] archives. Comparison between[...]"

We agree, we will modify the text

Page 12 – Line 21: Remove period after "BM"

We will modify the text accordingly

Page 17 – Line 31: Add superscript to 18 in $\delta$18O

We will modify the text accordingly

Page 24 – Line 8: italicize the latin "in situ"

We will modify the text accordingly

Page 24 (Fig. 6 caption): Add in what the whiskers represent (e.g. 95% confidence interval) and if the horizontal lines within the boxes are the median or mean. The lines are typically medians, but the text compares means of the datasets so you may want to show both the mean and the median (perhaps one as a bold line and one as a dashed line?)

We will expand the caption to include a note regarding the various components, the central bar of the boxplots is the median, the top and bottom of the box the 25th and 75th percentile, the whiskers are 1.5*IQR +/- the 75th or 25th percentile. The 95 % CI on the median is not shown here, as that is a box and whisker with 'notch' plot.

Page 26, 27, 28 (Fig captions): Note what "p($\delta$18O)" is. I initially thought it was a p-value.

We will modify the text accordingly

Page 10 - Line 43: Use parentheses only around the year, i.e. "Berger et al. (1978b)"

Page 11 - Line 5: Use parentheses only around the year, i.e. "Brummer et al. (1987, 1986)"

Page 11 - Line 6: Use parentheses only around the year, i.e. "Peeters et al. (1999)"

Page 11 - Line 27-29: Use parentheses only around the year, i.e. "Wit et al. (2010)".

Page 14 – Line 12: Remove the extra ", "

Page 17 – Line 24: Remove the extra ", "

We will correct these Endnote 'cite as you write' mistakes and modify the text accordingly

[Figure]

**Fig. 1.** Cross section of a shell of T. sacculifer

[revised manuscript text omitted]

experiment, we investigate if the oxygen isotope composition of shells of different species from the same geographic location share the same variability.

**1.2.1 Question 1: Do individuals belonging to the species *T. sacculifer* calcify at one specific depth, or undergo depth migration?**

5   The 'average' depth habitat of planktonic foraminifera of several species was first defined by (Emiliani, 1954) revealing that different species occupy discretely different depth habitats, independently corroborated by the later work of (Jones, 1967) by those same species presence and absence of species within the opening and closing nets. However, the offset in $\delta^{18}O$ measured between specimens growing within the euphotic surface waters and those collected from the seabed indicated that depth habitat is not confined to a single depth (Duplessy et al., 1981; Mix, 1987), instead this 'average' species depth habitat would be a

10  weighted average of the various chamber calcification depths occurring during an individual's ontogeny (Kozdon et al., 2009a; Kozdon et al., 2009b; Shuxi and Shackleton, 1989; Takagi et al., 2015, 2016). Data from plankton tow studies combined with reproduction at depth would suggest that foraminifera migrate through the water column during ontogeny (Figure 1). For certain species of foraminifera (*i.e.*, *T. sacculifer* and *G. ruber*), however, a portion of the shell may have grown deeper in the water column than the living depths estimated by plankton tows (Lohmann, 1995), *i.e.*, either a calcite crust triggered by

15  temperature change (Hemleben and Spindler, 1983; Hemleben et al., 1985; Srinivasan and Kennett, 1974) or reproduction triggered gametogenic calcification. For the first objective, we aim to test whether *T. sacculifer* performs depth migration, which would result in a deviation in the geochemistry between the different chambers of a single specimen and also in a deviation from the situation at the sea surface. A one-sample Student's t-test was used to test the claim that there is no difference between the mean of the final chamber and the remaining shell of *T. sacculifer*, *i.e.* the difference is equal to zero:

20  Let $X = \delta^{18}O_F - \delta^{18}O_{<F}$

$H_0$: $\mu_X = 0$ ,

$H_1$: $\mu_X \neq 0$ , (1)

By computing the difference and using a reference value of 0, we do not invalidate the rule of independence that a two-sample Student's t-test would require between the two sample populations. This dependence is based upon the inference that $\mu_F$ and

25  $\mu_{<F}$ could conceivably be considered to be 'before' and 'after' measurements and thus the value of $\mu_{<F}$ could have an impact upon the value of $\mu_F$.

**1.2.2 Question 2: Does the $\delta^{18}O_{shell}$ of *T. sacculifer* covary with size?**

Our second research objective is an expansion of the first objective, as deriving palaeo-SST from the $\delta^{18}O$ compositions of foraminiferal shell is based on the assumption that a given specimen calcifies at, or produces a large proportion of its shell at,

30  one specific depth in the water column. However, a portion of the variability associated with stable isotope measurements in foraminifera is believed to be size-dependent (Ezard et al., 2015). These size dependencies are typically attributed to biological effects and relate to depth migration through ontogeny (Feldmeijer et al., 2015; Metcalfe et al., 2015). For instance,

investigations into the population dynamics of living specimens of *T. sacculifer* in the central Red Sea revealed that whilst this species in general occupies the upper 80 meters of the water column distinct size classes were shown to have clear depth preferences (Bijma and Hemleben, 1994; Hemleben and Bijma, 1994) with small foraminifera (100 to 300 µm) in the upper 20 m and progressive larger foraminifera with depth to the point that the largest specimens (>700 µm) lived between 60 m and 80 m. Calcification at different depths throughout their life span may cause a deviation in the $\delta^{18}O$ values of individuals from different sizes, depending on the ambient water column structure, which would therefore reflect different depths and thus the selection of an appropriate size fraction may or may not unduly influence palaeoclimate reconstructions. The aim of this second objective is to further expand upon the results of our first question and test whether the different depth preferences for different sizes of *T. sacculifer*, have an effect on the $\delta^{18}O_{shell}$. Three size fractions were studied to learn more about the effect of size on $\delta^{18}O_{shell}$ and a one-way analysis of variance (One-way ANOVA) with a post-hoc test used to detect intra-sample differences was used to test the hypothesis that there is no difference between the means of the different size fractions of *T. sacculifer*:

$H_0: \mu_{small} = \mu_{medium} = \mu_{large}$ ,

$H_1$: at least one of the means is different, (2)

**1.2.3 Question 3: Do different species of planktic foraminifera from the same geographic location share the same single specimen $\delta^{18}O_{shell}$ variability?**

Having focused upon a single species for the first two research questions, our third question focuses upon the variability of foraminifera isotope values, which are considered to represent seasonality, and whether fossil shells from different species share similar $\delta^{18}O_{shell}$ variability. Commonly when referencing seasonality, temperature is considered as the variable of interest. However, the tropics have reduced temperature variation compared with higher latitudes, the core is situated along the northeast coast of Brazil which may be influenced by the shift in the ITCZ (Jaeschke et al., 2007). Temperature and salinity have opposing effects on the overall oxygen isotope composition. Surface dwelling species of planktonic foraminifera, *T. sacculifer* (Figure 2i: 2vii); *G. ruber* (Figure 2viii: 2xiii); and the thermocline dwelling *N. dutertrei* (Figure 2xiv: 2xix) were picked from the core top. All species are symbiotic (Schiebel and Hemleben, 2017) which limits the depth of the maximum growth. A One-Way ANOVA was used to test, whether the means of each species are equal or the if the alternative hypothesis that one or more of the species means differs from one another, with the following hypothesis:

$H_o: \mu_{T.sacculifer} = \mu_{G.ruber} = \mu_{N.dutertrei}$ ,

[revised manuscript text omitted]

---

## Author Response (AR2)

Dear Dr. de Nooijer,

On behalf of the co-authors please find our response to your comments (in green), corrections are shown as yellow highlight in the manuscript. We have, as per your instruction, altered figure 5.

Kind regards

Brett Metcalfe

page 7, line 17: Please avoid references to non-peer reviewed work (Peeters, unpublished data).

Citation removed

The significance of the trends in 5c and 5d is now explained at the end of section 3.2 (next time please indicate this explicitly in your reply to the reviewers). Following the second reviewer's comment, however, I agree that it would be informative to add the 95% CI's of the trendlines in 5c and 5d.

The lines and error bars have been added to both plots

Considering the second reviewer's comment on the vagueness of using "above and below 100 m" (again: in your reply please indicate where we can find the changes you made), I fail to see how your added text (end of 4.2) answers the question. Please respond more precisely by either narrowing down the depths or by explaining why you decided to refrain from it.

Giving a super-precise depth suggests that we know the exact season of the growth, that our calculation of equilibrium is perfect and that the measurements are without error. World Ocean Atlas is a 60 year Climatology, without any of the potential high frequency temperature variance we know exists on a day-to-day basis. Readers can clearly read off the temperature-depth relationship if they want that, but we are being vague on purpose so that someone doesn't read our paper and go T. sacculifer have a fixed depth of precisely Xm for their <F chambers. The original comment by the reviewer "above and below 100 m" is a bit vague. Note which water depths have temperatures of 24.5∘C and 25.5∘C at this site. If 25.5 ∘C measured from the <F fragment is not in the upper 0-50 m"  completely ignores the potential error associated with atlas / climatology data, it is not ocean reanalysis data.

It is very thoughtful of reviewer 2 to direct the authors to some unpublished results. I have no doubt that the Dr Wychech is correct in her statements on the contribution of GAM calcite to the overall oxygen isotopes, but it is (unfortunately) not ok to refer to unpublished results (and btw: the references are currently not in the revised version of the manuscript). Therefore, I suggest to remove the references and only add them if the relevant work has been published/ accepted in the meantime. The authors will therefore also need to rephrase the newly added text at the end of 4.2. I have no objection in keeping the general idea of the added text intact.

Unfortunately the reviewer's paper is submitted/in review at G³ so we have therefore changed the references to personal communication – we are unsure as to how to cite a reviewer comment in the discussion

Technical comments:
Abstract, lines 11-14. This sentence is difficult to understand. Please simplify or divide into several sentences.

We changed this to: In this study three different hypotheses were tested to gain more insight into biological and ecological processes that influence the resultant composition of stable isotopes of

oxygen ($\delta^{18}O$) in the shells of planktonic foraminifera. These hypotheses were related to: the shell size; the differences in isotopic composition between the final chamber and the remaining shell; and the differences between different species.

Abstract, line 16: please add "single" before "chamber".

Changed

Abstract, line 18: there is a verb missing here. Consider: "The formation of the final chamber happens at temperatures that are approximately ..., suggesting... ".

Changed

Abstract, lines 20-21: please remove "..., based upon measured size as opposed to sieve size, ...".

Changed

Introduction, page 2, line 5: replace "The knowledge of how" by "Understanding how".

Changed

Introduction, page 2, line 6: replace "in order" by "to".

Changed

Introduction, page 2, lines 6-7: replace "...reduce associated error. " by "...reduce the uncertainty in reconstructed parameters."

Changed

Introduction, page 2, line 15: should be "individuals' depth distribution".

Changed

I am not a big fan of the subdivision into 'Questions'. Consider renaming them by e.g. "depth migration by T. sacculifer" for Q1. Then the different methods sections can also be combined rather than be kept separate for every 'question' (these sections are often a bit short in the current version of the manuscript).

I agree with the Editor that the sections are relatively short, but I have to respectively disagree, we have three specific questions, we could give them 3 obtuse titles and put the methods into one section. But in our current format we are making sure that any reader skimming our paper knows that there are three separate methods for the three questions.

Introduction, page 3, lines 6-7: replace "by those same species... and closing nets." by "...by presences or absence of species in stratified net tows." or something similar.

Changed: by the presence or absence of species in stratified net tows

Introduction, page 3, line 18: replace 'situation' by 'conditions'

Changed

Introduction, page 4, line 9: "shell" should be subscript. Also noted at other places: please check the complete text for this typo.

Changed – also checked text

Introduction, page 4, line 19: "reduced temperature variation"? I suspect that this refers to a relatively small seasonal temperature variability. Please rephrase.

Changed to: However, the tropics have relatively small seasonal temperature variability compared with higher latitudes, the core is situated along the northeast coast of Brazil which may be influenced by the shift in the ITCZ

Introduction, page 4, line 30: distribution of what? del18O?

Changed

Methods, page 5, line 8: is there a reference for this protocol? Otherwise, remove "the methodological protocol of the VUA"

Changed
Methods, page 6, line 21: 'vital effect' is not (necessarily) the same as 'size effect': please replace the former by the latter.

We're not discussing a size effect correction but an actual vital effect correction. Changed for clarity :All $\delta^{18}O_{shell}$ values of *G. ruber* and *T. sacculifer*, irrespective of size, were corrected for their vital effect.
Methods, page 7, line 11-12: I don't understand this sentence, please rephrase.

Changed :Here, Kim and O'Neil (1997) is used to define an inorganic equilibrium value of $\delta^{18}O_c$ this equation is chosen to avoid potential differences due to (1) light-level; (2) foraminiferal size; (3) ontogenetic level (Bemis et al., 1998; Bemis et al., 2000); and (4) species (Mulitza et al., 1999b).
Methods, page 7, line 17: insert a comma after 'results'

Changed
Results, page 7, line 23: 'To aid the reader' is awkward phrasing. To aid (the reader) with what?

  ➢ One reviewer suggested that we reduce the number of decimal places, however this could lead to rounding errors in that statistics, to aid the reader we have left the results in a longer format.
  ➢ I don't exactly get how 'to aid the reader' is awkward phrasing in such a context.

Results, page 7, line 25: 'errors'.

Changed
Results, page 7, line 27-28: it is obvious that measuring size and weighing generates data about size and weight. Remove "…, generating data about size and weight".

Removed
Results, page 8, line 2: replace 'greater' by 'larger'.

Changed
Results, page 8, line 3: should be: "…or non-linear growth of the foraminiferal shell."

Changed
Results, page 8, lines 4-5: this is only true for Rotaliid foraminifera, not foraminifera in general.

Changed: After completion of the first chamber, during the construction of subsequent chambers of a Rotaliid foraminifer an additional layer of calcite is added to the previous chambers, making them a incrementally thicker.

Results, page 8, line 9: 'larger' instead of 'bigger'.

Changed
Results, page 8, line 9: what does 'wilder' mean?

  ➢ Changed to erractic

Results, page 8, line 11: 'F' should be 'F chamber' I guess.

Changed
Results, page 8, line 11: 'the shell' likely refers to the shell of large specimens. Please add.

Changed
Results, page 8, line 14: 'the area of the final chamber'.

Changed
Results, page 8, line 15: please remove '; values become more scattered".

Changed
Results, page 8, line 15: As shell 'sizes' increase?

Changed

Results, page 8, line 16: replace 'body' by 'shell'.

Changed
Results, page 8, line 16-17: this sentence is unclear. Please rephrase. Suggestion: "Large specimens often have disproportionally large or small final chambers."
Changed :Large specimens often have disproportionally large or small final chambers, with no clear relationship between total shell size and the size of the final chamber.

Results, page 8, line 24: try to avoid using 'we'.

Changed
Results, page 9, line 1: insert another comma before 'respectively'.

Changed
Results, page 9, lines 8-9: insert another comma before 'respectively'.

Changed

[revised manuscript text omitted]